# Geographical variation and predictors of missing essential newborn care items during the immediate postpartum period in Ethiopia: Spatial and multilevel count analyses

**Aklilu Habte Hailegebireal**[1]*, **Aiggan Tamene Kitila**[2,3]

**1** School of Public Health, College of Medicine and Health Sciences, Wachemo University, Hosanna, Ethiopia, **2** Centre for Sustainability, University of Otago, Dunedin, New Zealand, **3** Department of Preventive and Social Medicine, University of Otago, Dunedin, New Zealand

* akliluhabte57@gmail.com

**Data Availability Statement:** The data for this study were obtained from the DHS program with a reasonable request. Because the DHS office

## Abstract

### Background

Essential Newborn care (ENC) is a High-quality universal newborn health care devised by the World Health Organization for the provision of prompt interventions rendered to newborns during the postpartum period. Even though conducting comprehensive studies could provide a data-driven approach to tackling barriers to service adoption, there was a dearth of studies in Ethiopia that assess the geographical variation and predictors of missing ENC. Hence, this study aimed to identify geographical, individual, and community-level predictors of missing ENC messages at the national level.

### Methods

This study used the 2016 Ethiopian Demographic and Health Survey, by using a weighted sample of 7,590 women who gave birth within two years prior to the survey. The spatial analysis was carried out using Arc-GIS version 10.7 and SaTScan version 9.6 statistical software. Spatial autocorrelation (Moran's I) was checked to figure out the non-randomness of the spatial variation of missing ENC in Ethiopia. Six items of care used to construct a composite index.0of ENC uptake were cord examination, temperature measurement, counselling on danger signs, counselling on breastfeeding, observation of breastfeeding, and measurement of birth weight. To assess the presence of significant differences in the mean number of ENC items across covariates, independent t-tests and one-way ANOVA were performed. Finally, a multilevel multivariable mixed-effect negative binomial regression was done by using STATA version 16. The adjusted incidence rate ratio (aIRR) with its corresponding 95% CI was used as a measure of association and variables with a p-value<0.05 were identified as significant predictors of ENC.

doesn't allow to sharing of the data with other third parties, the one who needs the data supporting the findings of this study can get it in anonymized form from the DHS website at https://www.dhsprogram.com upon reasonable request. The authors did not have any special access privileges that others would not have.

**Funding:** The author(s) received no specific funding for this work.

**Competing interests:** The authors have declared that no competing interests exist.

**Abbreviations:** AIC, Akaike's information criterion; ANC, Antenatal Care; CSA, Central Statistical Agency; EDHS, Ethiopian Demographic and Health Survey; ENC, Essential Newborn Care; GWR, Geographically Weighted Regression; IRR, Incidence Rate Ratio; OLS, Ordinary Least Square; PNC, Postnatal Care; PPP, Postpartum Period; WHO, World Health Organization.

## Results

The overall prevalence of missing ENC was 4,675 (61.6%) (95% CI: 60.5, 62.7) with a significant spatial variation across regions. The majority of Somali, Afar, south Amhara, and SNNPR regions had statistically significant hotspots for missing ENC. The mean (±SD) number of ENC items received was 1.23(±1.74) with a variance of 3.02 indicating over-dispersion. Living in the poorest wealth quintile (aIRR = 0.67, 95%CI: 0.51, 0.87), lack of Antenatal care (aIRR = 0.52, 95%CI: 0.49, 0.71), birth at home (aIRR = 0.27, 95% CI: 0.17, 0.34), living in rural area (aIRR = 0.39, 95% CI: 0.24, 0.57) were significant predictors of ENC uptake.

## Conclusion

The level of missing ENC was found to be high in Ethiopia with a significant spatial variation across regions. Hence, the government and policymakers should devise strategies for hotspot areas to improve women's economic capabilities, access to education, and health-seeking behaviours for prenatal care and skilled delivery services to improve ENC uptake.

## Introduction

The neonatal period (the first month of life) is the most vulnerable and critical period for a child's survival, growth, and development [1]. Approximately one million babies died during the first 24 hours of life in 2019, accounting for almost three-quarters (75%) of all neonatal deaths globally [2]. About 2.4 million newborns died during the neonatal period in 2020, with approximately 6700 newborn deaths every day, which represented nearly half (47%) of all under-5 deaths [2]. Sub-Saharan Africa(SSA) and Central and Southern Asia(CSA), with neonatal mortality rates (NNM) of 27 and 23 per 1000 live births, respectively, accounted for 43% and 36% of global neonatal deaths [2, 3]. A child born in SSA is ten times more likely than a child born in a high-income country to die in the first month [2]. Ethiopia has had significant achievement in reducing under-5 mortality rates by 71% over the last two decades, from 204 deaths per 1,000 live births in 1990 to 59 deaths per 1,000 live births in the 2019 mini EDHS report [4, 5]. However, the decline in NNM from 58 deaths per 1000 live births to 29 deaths per 1000 live births between 2000 and 2016 has not been satisfactory [4]. Even so, it has risen to 33 deaths per 1,000 live births in 2019 [4].

The postpartum period (PPP), which begins immediately after delivery and lasts up to six weeks (42 days), is an important time for women, babies, partners, parents, carers, and families [6]. The majority of neonatal deaths in developing countries including Ethiopia were due to a lack of quality essential newborn care (ENC), especially within the first two days of birth [2]. Thus, the World Health Organization (WHO) recently devised global recommendations on the timing and contents of postnatal care (PNC) for mothers and newborns, in resource-limited settings to ensure a positive postnatal experience [6]. One of the basic elements of PNC aimed at improving newborn survival is the provision of easy, low-cost, and less time-consuming ENC in the immediate PPP [7].

Essential Newborn care (ENC) is a WHO strategic approach for the provision of a constellation of interventions rendered to newborns during the immediate PPP [7, 8]. It is a High-quality universal newborn health care that entails prompt care at the time of birth and throughout the neonatal period, regardless of the place of delivery [8, 9]. It comprises

measurement of body weight, temperature monitoring, hygienic cord care, early and exclusive breastfeeding, evaluation for danger signs, and preventive services like immunization [2, 7]. Findings showed that providing quality ENC is a cost-effective strategic approach that results in a considerable reduction in early and late neonatal death [10–12].

Despite its importance in the reduction of NNM, 29.3%, 32.3%, 35.5%, and 31% of women in Nepal [13], India [14], Rwanda [15], and Tanzania[16] missed ENC, respectively. However, a recent systematic review meta-analysis in Ethiopia revealed that the uptake of ENC services was found to be low at 48.77% [17]. Inadequate intrapartum (skilled delivery service) and post-partum care (like ENC) accounted for a steady reduction in neonatal deaths in Ethiopia [4, 18]. The government has been striving to meet the third Sustainable Development Goal (SDG3), which is aimed at ensuring healthy lives and promoting well-being for all people through the provision of adequate maternal and newborn health services [19]. ENC service delivery at the health facility and community level is one approach to meeting this ambitious goal [20].

Despite several studies on ENC uptake in Ethiopia, none investigated the spatial distribution of missing service usage and its predictors at the individual and community levels by using a large population. Even though a recent study attempted to examine the spatial distribution, it primarily focused on women who received ANC [9] and may have failed to provide a clear picture of ENC. Thus, the current study used a relatively larger sample size of women who were eligible for PNC and employed spatial (a hot spot and a geographically weighted regression (GWR) analysis) and multilevel count analytic approaches. A hotspot analysis was performed to identify regions with a high proportion of women who missed ENC, followed by a GWR to identify potential predictors that lead to regional disparities in missing the service. In addition, a multilevel approach was done to identify factors at the individual and community levels that contribute to missing ENC items. Conducting such comprehensive analyses could provide a data-driven approach to comprehending and reducing the barriers to service uptake [21]. The findings will enable stakeholders to strengthen their efforts in addressing the impediments to service uptake by designing geographical, individual, and community-focused effective interventions to ensure a positive postnatal experience.

## Methods and materials

### Data source, study design, and period

This study used the 2016 Ethiopian Demographic and Health Survey (EDHS) report; a population-based, nationally representative survey collected from January 18 to June 27, 2016. The country is located at $3^0$–$15^0$ N latitude and $33^0$–$48^0$ E longitude. The country has a total surface area of 1,112,000 km2 and has boundaries defined on the west by Sudan, on the east by Somali and Djibouti, on the north by Eritrea, and on the south by Kenya. Nine regions (Tigray, Afar, Amhara, Oromia, Somali, Benishangul-Gumuz, Southern Nation Nationality and People's Region (SNNPR), Gambella, and Harari) and two self-administrative cities (Addis Ababa and Dire Dawa) were included in the survey [22]. The data were obtained from the women's (IR) file contained in the 2016 EDHS report.

### Population of the study

The source populations were all women who responded to a query about receiving a postnatal check-up within two months. The study population, on the other hand, were mothers who had complete information on the uptake of each essential newborn care within the first two days following delivery. A total of 8,490 respondents were excluded from the study due to a lack of information on service usage (missing values).

## Sampling procedure and data collection tools

Study participants were selected through a stratified two-stage cluster sampling technique, where each region was divided into urban and rural areas. In the first stage, 645 clusters or enumeration areas (202 urban and 443 rural areas) were selected randomly. Then, a fixed number of 28 households with eligible women were selected per cluster based on an equal probability systematic selection. The survey design and methodology were addressed as well in the 2016 EDHS [22]. The Global Positioning System (GPS) was used to obtain the geographic coordinates of each survey cluster [22, 23]. To assure the confidentiality of respondents, the geographical locations(latitude and longitude) were randomly displaced. The greatest displacement was two kilometers (km) for all urban and five kilometers (km) for 99% of rural clusters. The remaining 1% of rural clusters have been displaced up to a 10-kilometer distance [24]. Data were collected from all eligible women using the Woman's Questionnaire, which includes socio-demographic and economic information, obstetric characteristics, and maternal health service usage.

## Measurement of variables of the study

**Outcome variable.**   The outcome variable for this study was the uptake of essential newborn care services during the immediate postpartum period (PPP). The measurement was based on the receipt of six essential services: (i) cord examination (m78a_1), (ii) Temperature measurement (m78b_1), (iii.) Counselling on danger signs (m78c_1) (iv) Counselling on breastfeeding (m78d_1), (v) Observation of breastfeeding (m78e_1), and measurement of birth weight (m19_1). There were yes, and no response options for each question. For the sake of analysis, the categories were confined to Yes (= 1) and No (= 0). A composite index of essential newborn care (ENC) has been created based on the responses, which is a count of the number. The variable had a minimum and maximum value of zero and six, respectively. Finally, women with a value of '0' were considered as 'missing ENC service,' whilst those who scored one or more were considered to have 'received ENC', and the spatial analyses were focused on those who did not receive the service [9, 25].

**Explanatory variables.**   After reviewing related and current literature, potential predictors of ENC were selected from the data set and nested at the individual and community levels [7, 25–27] (Table 1).

## Data management and statistical analysis

STATA version 16, ArcGIS version 10.7, and SaTScan version 9.6 were used to analyze the data. The data were weighted to minimize under- or over-representation during strata-specific selection during the survey. This is vital to obtain a reliable estimate and draw proper inferences[30]. The weighted proportions of an ENC and its predictors were computed in STATA and prepared in Microsoft Excel 2016 (CSV format) before being loaded into ArcGIS 10.7 for further spatial analysis. Descriptive statistics such as frequency and percentage of different variables were estimated and displayed using texts, and tables.

**Spatial analyses.**   Initially, the data containing the variable of interest in CSV format were imported into ArcGIS version 10.8 and combined with the GPS data (shape file). After that, the event data was converted to a shape file and displayed in an XY plane (geographically coordinated system). Projections of the geographically coordinated data to the projected coordinate data were performed before the analysis.

**Spatial autocorrelation (Global Moran's I).**   The global spatial autocorrelation was estimated using Moran's index to decide whether missing ENC in Ethiopia was dispersed, clustered, or randomly distributed. In general, a Moran's I value close to 1 suggests significant

**Table 1. List of possible predictors of missing essential newborn care items during the immediate postpartum period in Ethiopia as extracted from the EDHS 2016 report.**

| Variables | Description | Category |
|---|---|---|
| Age | The respondent's age, expressed in years, at the time of the survey. | 1. 15–19, <br> 2. 20–34 <br> 3. 35–49* |
| Marital status | Percentage of women according to the current status of marriage or cohabitation. | 1. In marital relationships* <br> 2. Not in a marital relationship |
| Level of Education | Percent distribution of women ages 15–49 by the highest level of schooling attended or completed. | 1. No education <br> 2. Primary <br> 3. Secondary/higher* |
| Family size | Number of household members at the time of data collection | 1. ≤5* <br> 2. >5 |
| Sex of head of Household | Percent distribution of households by sex of head of household | 1. Female <br> 2. Male |
| Wealth index | Calculated using straightforward information on a household's ownership of certain goods, such as televisions and bicycles; housing materials; livestock, crop production, and access to water, sanitation, and hygiene | 1. Richest* <br> 2. Richer <br> 3. Middle <br> 4. Poorer <br> 5. Poorest |
| Parity | The number of living children the woman had at the time of the survey | 1. Nulliparous <br> 2. Primiparous <br> 3. Multiparous <br> 4. Grand multiparous* |
| Antenatal care | Number of women who received antenatal care for their last birth, which was initially reported in continuous form and then grouped as no antenatal care, 1 visit, 2–3 visits, 4+ visits | 1. No ANC visit <br> 2. One visit <br> 3. Two-three visits <br> 4. Four and more visits* |
| Pregnancy status during last childbirth | Percentage of births to women aged 15 to 49 in the five years preceding the survey, including current pregnancies, by planning status of the last pregnancy—(i) wanted then, (ii) wanted later, or (iii) not wanted at all. | Unwanted (*ii& iii*) <br> Wanted (*i*) * |
| Place of delivery | Percent distribution of live births in the past 5 years by place of delivery. | 1. Health facility* <br> 2. Home |
| Media exposure | The number of women aged 15 to 49 who are exposed to specific media at various frequencies, such as reading a newspaper, watching television, and listening to the radio. | 1. Not at all <br> 2. Less than once a week <br> 3. At least once a week* |
| Autonomy in decision-making [a] | The total number of married women between the ages of 15 and 49 who make decisions for their own health care, major household buys, and visits to family or relatives. | 1. Low <br> 2. high* |
| | **Community level factors** | |
| Residence | The area where respondents lived when the survey was conducted. | 1. Urban* <br> 2. Rural |

(*Continued*)

**Table 1.** (Continued)

| Variables | Description | Category |
|---|---|---|
| Region | The geographically delineated area where the woman was resided at the time of the survey. Three categories were created as Small periphery regions (Afar, Somali, Benishangul, and Gambella), Major central regions (SNNPRs, Tigray, Amhara, and Oromia) and Metropolitans (Addis Ababa, Dire Dawa, and the Harari region) | 1. Small periphery<br>2. Major central regions<br>3. Metropolitans* |

*reference category

ᵃ Autonomy in decision-making: was assessed by using three questions about who makes the final decision for the family on large property purchases, visits to relatives, and health care. The response categories were (i) woman alone, (ii) woman and husband/partner, (iii) husband/partner alone, (iv) someone else, and (v) others. For each question, responses (i) or (ii) got a score of 1, indicating good decision-making capacity, whereas the remaining responses received a score of 0, indicating weak decision-making capacity. All the responses were summed to yield an overall score ranging from 0 to 3. Finally, a composite score had been divided into two distinct groups: low and high for "0 to 2" and "3" scores [28, 29].

positive autocorrelation (missing ENC was clustered/non-random), whereas a Moran's I value close to -1 shows significant negative autocorrelation (missing ENC was dispersed) across enumeration areas (EAs). On the other hand, the value near zero implies that the spatial distribution of missing ENC was random (independence between EAs or no spatial autocorrelation) [31, 32]. Accordingly, Moran's I value was statistically significant (p<0.05) which indicates the spatial distribution for missing ENC is non-random (clustered) [32].

**Spatial interpolation.** The spatial interpolation method is used to predict the likelihood of missing ENC in unsampled areas based on sampled EAs. There are multiple geostatistical and deterministic interpolation techniques, and for this study, ordinary Kriging was performed since it optimizes the weight and has a low residual and mean square error [33, 34].

**The spatial scan statistical (SaTScan) analysis.** A SaTScan analysis was carried out to show significant hot spots of missing ENC. The Bernoulli-based model was employed to detect the significant spatial clusters with no ENC since it uses a scanning window that moves across the study area [35]. To fit the Bernoulli model, women who missed ENC were treated as cases, while those who received the service were treated as controls. SaTScan statistics scanned gradually across the space to identify the number of observed and expected observations inside the window at each location. The default maximum spatial cluster size of 50% of the population was used as an upper limit, allowing both tiny and big clusters to be recognized. Using 999 Monte Carlo replications, the primary, secondary, and other significant clusters were identified and ranked based on the likelihood ratio test (LLR). A potential cluster, in the scanning window with the highest LLR and a significant p-value was selected as the high-performing cluster for being a case (missing ENC).

**Hot spot analysis (Gettis-Ord Gi* statistics).** Getis-Ord Gi* statistics were carried out to identify significant hot spot and cold spot areas for missing ENC. The z-score was estimated to figure out the statistical significance of clustering, and the level of significance was set at p-value<0.05 with a 95% CI. Cold spot and hot spot were declared at z-score was less than -1.96 and greater than +1.96, respectively [36, 37].

**Spatial regression analysis.** Spatial regression analysis incorporates global (ordinary list squares) and local (geographically weighted regression) techniques [38, 39].

**Ordinary least square (OLS) analysis.** The Ordinary Least Squares (OLS) regression model is a global model that predicts only one coefficient per independent variable throughout the whole study area. The compliance of all assumptions was assessed to ensure the reliability of the findings in this model. First, the coefficients of the intercept and predictors have to be statistically significant having a positive or negative sign. Also, the presence of multicollinearity

among explanatory variables should be confirmed by examining their variance inflation factors (VIF), and variables with VIF > 10 were considered multicollinear and iteratively excluded from the model [40, 41]. Furthermore, the Koenker Bp statistic was used to determine if the model could be employed for a geographically weighted regression (GWR) analysis. In the current study, the Koenker statistics were significant (p-value<0.001), and GWR analysis was required to examine the distribution of key variables.

**Geographically weighted regression (GWR).** Unlike OLS, which fits a single linear regression equation to all of the clustered data, GWR selects data from neighbouring features, therefore the GWR coefficient has different values for each cluster [42]. Variables with p-values<0.05 in the OLS model were selected for GWR and discussed based on their coefficients.

**Multilevel generalized linear model (GLM).** Due to the hierarchical nature of the EDHS data, where women were nested inside families and households were nested within clusters, ordinary one-level regression models were not appropriate, and hence multilevel modelling was used. Carrying out a multilevel analysis for such hierarchical data allows us to minimize biased parameter estimation [43].

Because the number of ENC items received is a non-negative integer (count), most recent thinking in the field suggests using GLM models with a Poisson link as a model of choice [44–46]. However, Poisson regression has an assumption, called the assumption of equidispersion, requiring the variance of the count response variable should be equal to its mean [45, 46]. The mean and variance of the count outcome variable in the current study were 1.23 and 3.02, respectively, indicating the presence of over-dispersed. Thus, a multilevel mixed-effect negative binomial regression model is appropriate for such an over-dispersed outcome variable [46–48].

**Model building and selection.** *Fixed effects*. To look into the presence of statistically significant differences in the mean number of ENC items across each categorical variable, independent t-tests and one-way analysis of variance (ANOVA) were performed. Those variables with p-values<0.05 were incorporated into a multilevel negative binomial regression, where significant predictors of ENC were determined. Finally, the incident rate ratio (IRR) with a 95% confidence interval was reported, and statistical significance was determined at a p-value<0.05.

*Random effects*. Four distinct models were fitted using a multilevel approach. Model one (null model) is devoid of any explanatory variables. The second and third models comprised solely individual and community-level characteristics. The fourth (full) model included and accounted for all factors at the individual and community level factors. The Intraclass Correlation Coefficient (ICC), Median Odds Ratio (MOR), and Proportional Change in Variance (PCV) were estimated to measure the random effects (variability in ENC uptake between and across clusters).

ICC quantifies the degree of heterogeneity of ENC between clusters and estimated as:

$$ICC = \frac{\text{var(b)}}{\text{Var(b)} + \text{Var(w)}};$$

Where Var(b) is the variance at the group level and Var(w) refers to the variance of the distribution used in the negative binomial link function, which is $\pi 2/3 \approx 3.29$.

The proportion of change in variance (PCV) measures the proportion of the total observed individual variation that is explained by the between-cluster variations and estimated as

$$PCV = \frac{(\text{Va} - \text{Vb})}{\text{Va}} * 100,$$

where, $V_a$ is the variance of the initial model (null model), and $V_b$ = variance of the subsequent models (models 2, 3, and 4). The value lies between 0 and 1 ($0 < PCV < 1$) or 0 and 100% ($0 < PCV < 100\%$).

The median odds ratio (MOR) represents the change in odds of missing ENC items when moving from one cluster to another while holding individual-level predictors constant.

*Model fitness*. Deviance = -2 * (Log Likelihood (LL), Schwarz's Bayesian Information Criterion (BIC), and Akaike's information criterion (AIC) were used to determine the best model. Finally, the fourth model with the lowest deviance, AIC, and BIC values was chosen as the best-fit model for the current study (Table 5).

*Ethical consideration and consent to participate*. Following registration at the DHS Programme website with possible justification, ICF International provided written permission to use both the DHS and GPS datasets. The data obtained were used only for the registered research and were not shared with anyone other than the co-authors. The DHS also declared that informed consent was obtained from all subjects and/or their legal guardian during the primary data collection. Furthermore, as this is secondary data, the Institutional Review Board (IRB) of Wachemo University College of Medicine and Health Sciences declared that no formal ethics approval was required. However, the IRB ensured that the research approach was ethically compliant with national and international standards.

## Results

### Background characteristics of the respondents

The findings of this study were based on a total weighted sample of 7,590 women. The mean (±SD) age of women was 29.25 (±6.84), with over half (50.4%) of them belonging to the age group of 25–34 years. Three-quarters (75.8%) of women in the poorest wealth quintile and 85.5% of women without ANC missed ENC services. In addition, 68.2% of rural women and 72.0% of women with no formal education missed ENC service. Addis Ababa and Afar regions exhibited the highest (3.7) and lowest (0.41) mean number of ENC contents, respectively. In addition, there were statistically significant differences in the mean number of ENC services received across the educational level, wealth index, residence, region, place of delivery, and media exposure ($p < 0.001$) (Table 2).

### The level of missing ENC items and disparities across the regions

More than half, 61.6% (95% CI: 60.5, 62.7) of women missed at least one ENC item. The highest proportion of women who missed ENC was recorded in Oromia (73.6%), closely followed by Somali (73.1%) and Afar (72.7%) regions. The top three ENC items that the majority of women failed to receive were cord checks (90.5%), counselling about danger signs (89.3%), and temperature measurement (87.0%) (Table 3).

### Spatial analysis results

**Spatial distribution of missing ENC among respondents.**  Somali, Afar, Western Oromia, and Gambella had a larger proportion of neonates who did not receive ENC. Tigray, Addis Ababa, and the Eastern border of Amhara, on the other hand, had a low rate of missing ENC (Fig 1).

**Spatial autocorrelation of missing ENC.**  The global spatial autocorrelation analysis revealed that the spatial distribution of missing ENC was non-random (i.e. there was significant spatial variation) across the country (global Moran's I = 0.49, $p < 0.001$). The clustered

**Table 2. Overall proportion of missing ENC and mean distribution of receiving ENC items across different characteristics of women in Ethiopia, EDHS 2016.**

| Variable categories | Total [Weighted frequency (%)] | Missed ENC | | Mean number of ENC items received | |
|---|---|---|---|---|---|
| | | n(%) | p-value | Mean (95% CI) | p-value |
| **Current age** | | | | | |
| 15–24 | 1,804(23.8) | 1,021(56.6) | 0.036 | 1.23(1.15, 1.30) | 0.021[a] |
| 25–34 | 3,826(50.4) | 2,307(60.3) | | 1.30(1.24, 1.36) | |
| 35–49 | 1,959(25.8) | 1,347(68.8) | | 1.09(1.01, 1.16) | |
| **Regions** | | | | | |
| Major central regions | 6,899(90.9) | 4,354(63.1) | <0.001 | 1.18(1.12, 1.23) | <0.001[a] |
| Peripheral | 441(5.81) | 299(67.7) | | 0.73(0.68, 0.78) | |
| Metropolitans | 249(3.3) | 23(9.1) | | 2.46(2.34, 2.58) | |
| **Religion** | | | | | |
| Orthodox | 2,882(38.0) | 1,465(50.8) | <0.001 | 1.81(1.73, 1.89) | 0.052[a] |
| Muslim | 2,824(37.2) | 1,998(70.7) | | 0.93(0.88, 0.98) | |
| Protestant | 1,651(21.8) | 1,034(62.4 | | 1.02(0.94, 1.11) | |
| Catholic | 71(0.9) | 48(67.4) | | 1.08(.63, 1.53) | |
| Traditional | 96(1.3) | 74(76.8) | | 0.26(0.13, 0.39) | |
| Others | 64(0.8) | 57(89.3) | | 0.31(0.23, 0.43) | |
| **Marital status** | | | | | |
| In marital relationship | 7,020(92.5) | 4,343(61.9) | 0.032 | 1.20(1.15, 1.24) | 0.061[b] |
| Not in marital relationship | 570(7.5) | 333(58.4) | | 1.56(1.41, 1.71) | |
| **Residence** | | | | | |
| Urban | 969 (12.8) | 157(16.2) | <0.001 | 2.62(2.52, 2.72) | <0.001[b] |
| Rural | 6,621(87.2) | 4,519(68.2) | | 0.86(0.82, 0.89) | |
| **Wealth index combined** | | | | | |
| Poorest | 1,651(21.8) | 1,252(75.8) | 0.001 | 0.48(0.44, 0.52) | <0.001[a] |
| Poorer | 1,654(21.8) | 1,188(71.8) | | 0.91(0.83, 0.99) | |
| Middle | 1,588 (20.9) | 1,036(65.2) | | 1.11(1.01, 1.21) | |
| Richer | 1,426 (18.8) | 879(61.6) | | 1.26(1.15, 1.38) | |
| Richest | 1,269(16.7) | 321(25.3) | | 2.62(2.52, 2.71) | |
| **Educational status** | | | | | |
| No education | 4,791(63.1) | 3,449(72.0) | 0.001 | 0.75(0.71, 0.79) | <0.001[a] |
| Primary | 2,150(28.3) | 1,142(53.1) | | 1.61(1.53, 1.69) | |
| Secondary and higher | 649(8.6) | 85(13.1) | | 2.74(2.61, 2.86) | |
| **Head of household** | | | | | |
| Male | 6,474(85.3) | 4,059(62.7) | 0.003 | 1.19(1.14, 1.23) | 0.991[b] |
| Female | 1,116(14.7) | 617(55.3) | | 1.36(1.27, 1.45) | |
| **Parity** | | | | | |
| Nulliparous | 49(0.7) | 33(67.1) | <0.001 | 0.68(0.37, 0.99) | 0.034[a] |
| Primiparous | 1,536(20.2) | 688(44.8) | | 1.75(1.66, 1.84) | |
| Multiparous | 3,478(45.8) | 2,128(61.2) | | 1.27 (1.21, 1.33) | |
| Grand multiparous | 2,527(33.3) | 1,827(72.3) | | 0.83(0.76, 0.88) | |
| **Frequency of ANC** | | | | | |
| No visit | 2,833(37.3) | 2,421(85.5) | <0.001 | 0.32(0.29, 0.35) | <0.001[a] |
| One visit | 334(4.4) | 225(67.4) | | 0.96(0.81, 1.12) | |
| 2–3 visits | 2,007 (26.5) | 1,124(56.0) | | 1.20(1.13, 1.28) | |
| ≥4 visits | 2,415(31.8) | 905(37.5) | | 2.15 (2.08, 2.23) | |
| **Place of delivery** | | | | | |

(*Continued*)

**Table 2.** (Continued)

| Variable categories | Total [Weighted frequency (%)] | Missed ENC | | Mean number of ENC items received | |
|---|---|---|---|---|---|
| | | n(%) | p-value | Mean (95% CI) | p-value |
| Home | 5,066(66.7) | 4,217(83.2) | <0.001 | 0.35(0.32, 0.37) | <0.001[b] |
| Health facilities | 2,523(33.3) | 459(18.2) | | 2.61(2.54, 2.68) | |
| **Planning status of pregnancy** | | | | | |
| Wanted | 5,573(73.4) | 3,377(39.4) | 0.206 | 1.22 (1.17, 1.26) | 0.820[b] |
| Unwanted | 2,016(26.6) | 1,298(64.4) | | 1.27(1.18, 1.35) | |
| **Ever terminate pregnancy** | | | | | |
| Yes | 680(8.7) | 427(62.8) | 0.636 | 1.28(1.14, 1.41) | 0.448 |
| No | 6,909(91.3) | 4,248(61.5) | | 1.22(1.18, 1.26) | |
| **Listen to radio** | | | | | |
| Not at all | 5,491(72.3) | 3,679(67.0) | <0.001 | 0.94(0.90, 0.98) | 0.031[a] |
| Less than once a week | 1,030(13.6) | 503(48.9) | | 1.89(1.77, 2.02) | |
| At least once a week | 1,069(14.1) | 494(46.2) | | 2.23(2.09, 2.36) | |
| **Watching TV** | | | | | |
| Not at all | 6,102(68.4) | 4,176(68.4) | <0.001 | 0.85(0.81, 0.89) | <0.001[a] |
| Less than once a week | 764(10.0) | 373(48.8) | | 1.71 (1.57, 1.86) | |
| At least once a week | 724(9.5) | 127(17.5) | | 2.90(2.78, 3.02) | |
| **Reading newspaper** | | | | | |
| Not at all | 7,050(92.9) | 4,566(64.8) | 0.063 | 1.07(1.03, 1.11) | 0.022 [a] |
| Less than once a week | 404(5.3) | 79(19.6) | | 2.98(2.80, 3.16) | |
| At least once a week | 135(1.8) | 30(22.5) | | 2.86 (2.54, 3.19) | |
| **Own mobile phone** | | | | | |
| Yes | 1,373(18.1) | 446(32.4) | <0.001 | 2.34(2.25, 2.43) | <0.001[b] |
| No | 6,216(81.9) | 4,230(68.0) | | 0.83(0.79, 0.87) | |
| **Covered by Health Insurance** | | | | | |
| Yes | 317 (95.8) | 152(47.8) | p<0.001 | 2.08(1.83, 2.32) | <0.001[b] |
| No | 7,272(4.2) | 4,524(62.2) | | 1.19(1.16, 1.24) | |
| **Autonomy in decision making** | | | | | |
| Autonomous | 5,395() | 3,232(59.9) | <0.001 | 1.31(1.26, 1.36) | 0.031[b] |
| Non-autonomous | 2,194() | 1,443(65.8) | | 1.02(0.95, 1.09) | |
| **Ease of distance to seek medical care** | | | | | |
| Big problem | 4,406(58.0) | 3,111(70.6) | <0.001 | 0.82(0.77, 0.86) | <0.001[b] |
| Not a big problem | 3,183(42.0) | 1,565(49.2) | | 1.69(1.63, 1.75) | |
| **Access to money for seeking medical care** | | | | | |
| Big problem | 4,547(59.9) | 3,096(68.1) | 0.023 | 0.92(0.87, 0.96) | <0.001 |
| Not a big problem | 3,043(40.1) | 1,580(51.9) | | 1.64(1.57, 1.70) | |

[a] p-values are based on an independent t-test

[b] p-values are based on a one-way Analysis of variance (ANOVA)

patterns (on the right sides) suggest that missing ENC occurred at a high rate throughout the study area (Fig 2).

**Incremental autocorrelation.** To determine the average nearest neighbour and minimum and maximum distance band, the incremental spatial autocorrelation over a series of distances depicted by a line graph with a corresponding z-score was performed. With a starting distance of 121803 meters, a total of 10 distance bands were found, with the first highest peak (clustering) observed at 151367.66 meters (Fig 3).

**Table 3. The level of missing overall ENC and contents of care across regions of Ethiopia, EDHS 2016.**

| Regions | Women who missed each ENC item [Weighted frequency (%)] | | | | | | |
|---|---|---|---|---|---|---|---|
| | Birth weight measurement | Checking cord | Temperature measurement | Counselling on danger sign | Counselling on breastfeeding | Observing breastfeeding | Missing all items |
| Tigray | 359(66.8) | 384(71.5) | 357(66.5) | 378(70.3) | 252 (46.9) | 207(38.6) | 139(25.9) |
| Afar | 66(92.3) | 69(97.0) | 69(96.8) | 67(94.9) | 61(86.4) | 55 (77.9) | 52(72.7) |
| Amhara | 1,448(88.7) | 1512(92.6) | 1,407(86.2) | 1,482(90.8) | 1,145(70.2) | 1,149 (70.4) | 971(59.5) |
| Oromia | 2,770(88.5) | 2909(92.9) | 2,872(91.8) | 2,954(94.4) | 2,683(85.7) | 2,603(89.2) | 2,303(73.6) |
| Somali | 236(87.7) | 255(94.9) | 253(94.1) | 261(97.3) | 234(87.3) | 226 (84.2) | 197(73.1) |
| Benishangul | 60(74.0) | 74(91.8) | 71(87.8) | 72(88.8) | 57(71.2) | 47(58.4) | 41(50.4) |
| SNNPR | 1,331(83.2) | 1477(92.3) | 1,430(89.3) | 1,386(86.6) | 1,176(73.5) | 1,160(72.5) | 940(58.8) |
| Gambella | 14(65.1) | 19(93.9) | 19(91.3) | 19(93.0) | 16(78.6) | 16(76.9) | 10(47.1) |
| Harari | 10(56.3) | 14(78.7) | 14(80.3) | 15(84.1) | 12(66.0) | 8(43.8) | 5(29.1) |
| Addis Ababa | 22(11.0) | 125(63.3) | 89(44.7) | 116(58.8) | 52(26.4) | 53(27.0) | 7(3.6) |
| Dire Dawa | 16(48.6) | 27(80.8) | 26(79.6) | 28(83.1) | 19(59.6) | 22(65.8) | 10(31.1) |
| **Total** | 6,331(83.4) | 6867(90.5) | 6,607(87.0) | 6,778(89.3) | 5,712(75.2) | 5,549(73.1) | 4,675(61.6) |

**Fig 1. Spatial distribution of missing ENC in Ethiopia, EDHS 2016.**

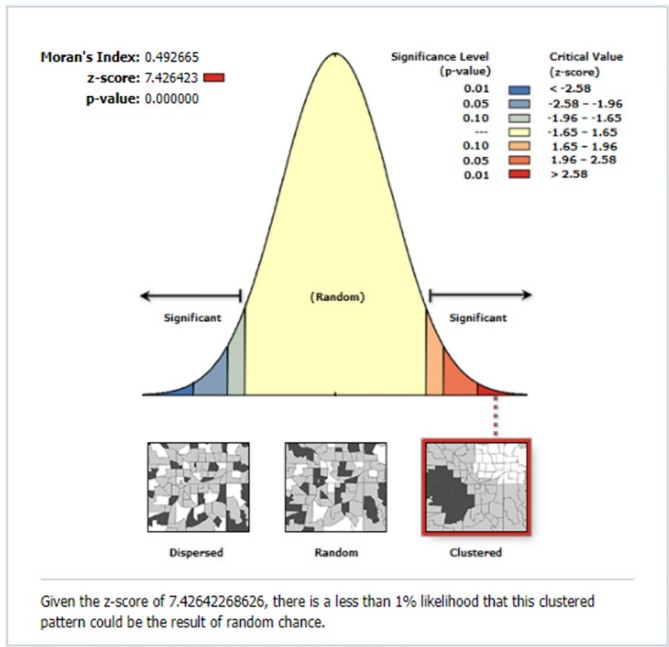

**Fig 2. The global spatial autocorrelation of missing ENC in Ethiopia, EDHS 2016.**

**Hot spot (Getis-Ord Gi\*) analysis.** As evidenced by hot spot analysis, Somali, central and southwest Afar, the southern part of Amhara, southwest Oromia, and the north and eastern portions of SNNPR have a high rate of missing ENC. Tigray, Addis Ababa, and Dire Dawa, on the other hand, had a low percentage of missing ENC (Fig 4).

**Spatial interpolation.** The spatial distribution of missing ENC for places where data were not collected was estimated using the ordinary Kriging technique. The highest predicted prevalence of missing ENC(red-shaded) was found in southern Somali, North and South Afar, Southwest and central Oromia, and parts of Addis Ababa and Gambella. In contrast, the expected proportion of high ENC uptake (green-shaded) includes the entire Tigray region, and central parts of Addis Ababa, Gambella, Dire Dawa, and Harari (Fig 5).

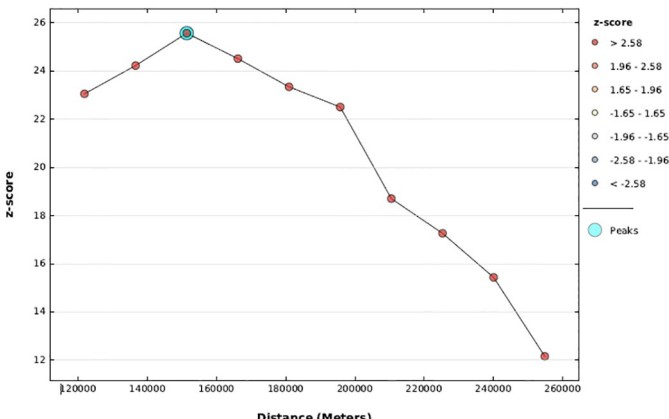

**Fig 3. The incremental autocorrelation of missing ENC in Ethiopia, EDHS 2016.**

## Hot Spot and Cold spot of missing newborn care during immediate postpartum period in Ethiopia

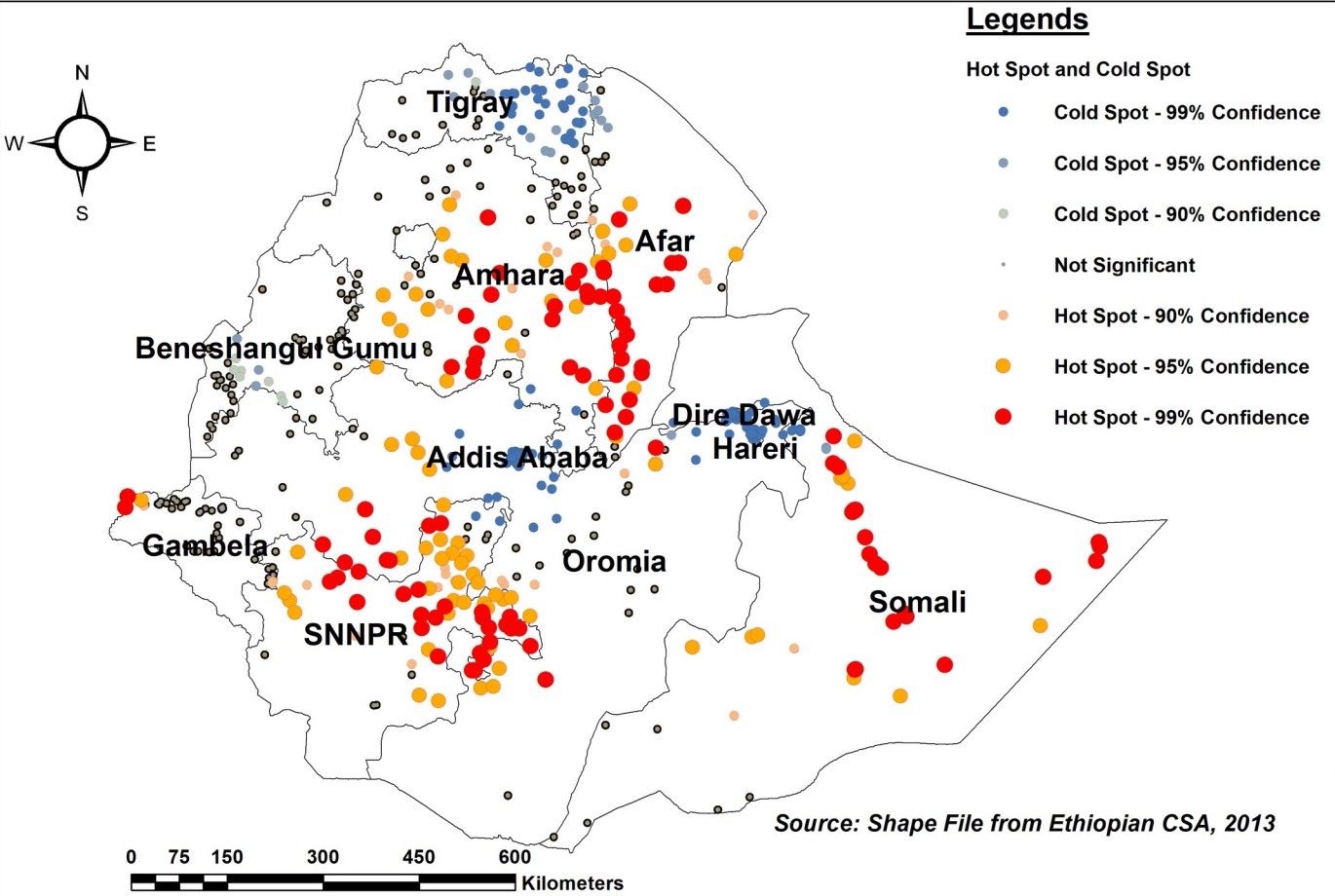

**Fig 4. Hot spot and Cold spot analysis of missing ENC across regions in Ethiopia, EDHS 2016.**

**Spatial scan statistical (SaTScan) analysis.** The SaTScan spatial analysis identified fifteen statistically significant groups of SaTScan clusters with a high proportion of newborns missing ENC. This means that the prevalence of missing ENC was higher inside the SaTScan circular window than outside of it. In addition, the analysis identified a total of 333 significant clusters that accounted for missing ENC, of which 78 and 42 were found in the first and second most likely clusters respectively. The first most likely cluster located at geographical coordinates of (5.330795 N, 41.837597 E) with a 427.4 Km radius, and LLR of 82.86 at p<0.001 showed that newborn within the area had a 41% (RR = 1.41) higher risk to miss ENC than their counterparts outside the area (Table 4).

## Results of spatial regression

**The global ordinary least square (OLS) analysis results.** The OLS model is the first step toward choosing the appropriate predictors for the spatial variation of missing ENC. As a result, lack of formal education, giving birth at home, lacking ANC, and never watching television were found to be associated with missing ENC. There was no sign of multicollinearity

Interpolated Spatial distribution of not receiveing essential newborn care during immediate postpartum period in ethiopia

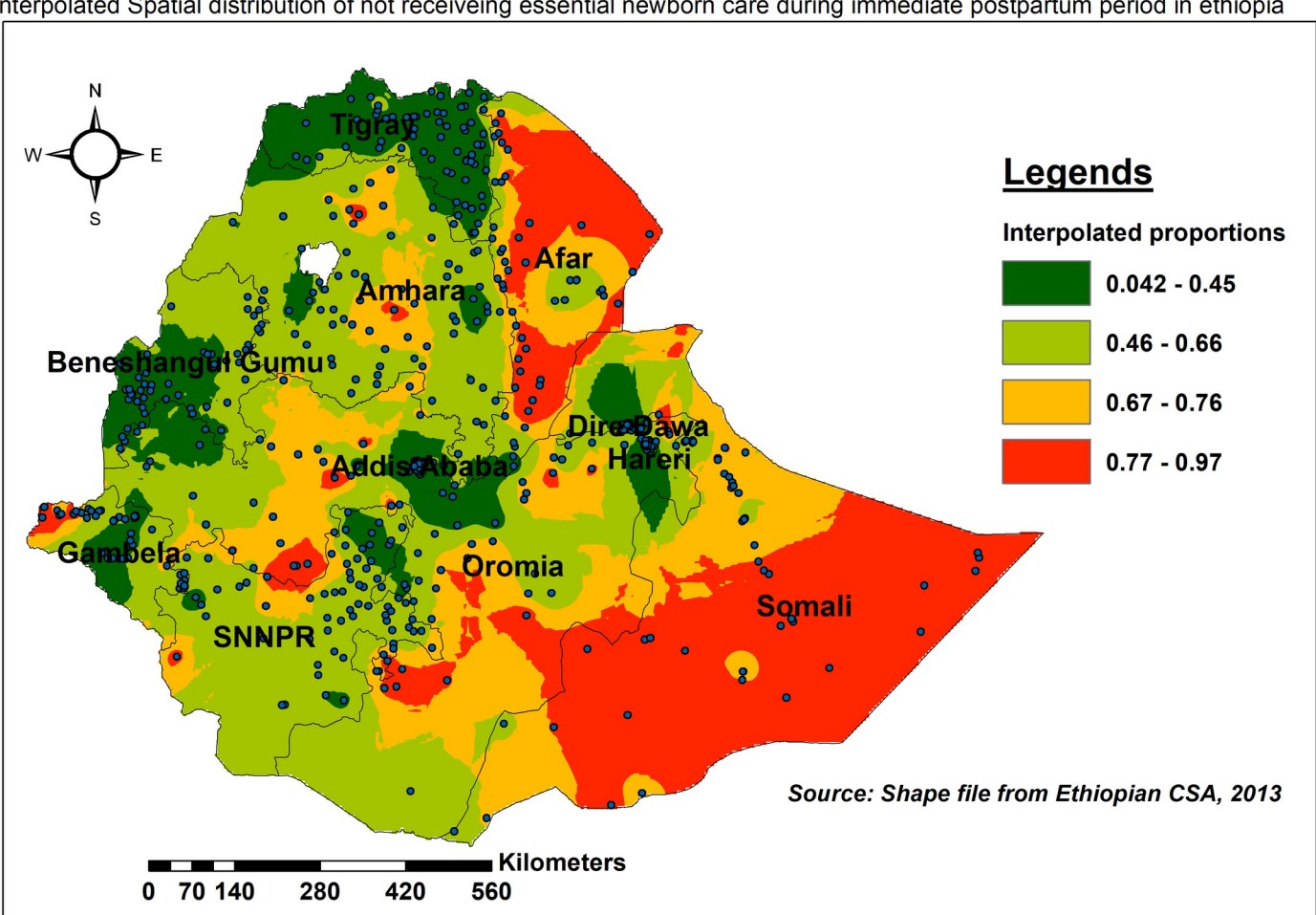

**Fig 5. Ordinary Kriging interpolation of the spatial distribution of missing ENC in Ethiopia, EDHS 2016.**

among the selected predictors (mean VIF = 1.40, minimum VIF = 1.19, and maximum VIF = 1.94). Furthermore, the adjusted $R^2$ = 0.718 from the OLS global model output indicated that the four predictors explained 71.8% of the variation in missing ENC. Jarque-Bera statistics with p-value>0.05 (p = 0.448) suggest that the model prediction was not biased (the requirement for residual normality was met). The Koenker statistics in the model, on the other hand, showed a statistically significant p-value (p<0.001), showing that the regression model is inconsistent across the study area, implying the necessity for the GWR model to estimate the model parameters properly (Table 5).

## Geographically weighted regression analysis

To deal with this violation of the stationarity assumption of the global (OLS) model, the local (GWR) model was fitted to offer realistic estimates. The GWR analysis outperformed the global model (OLS) significantly. The AICc value in the GWR model reduced from -349.68 in the OLS model to -401.11. The adjusted $R^2$ in GWR was greater than the OLS one, showing

**Table 4. The most likely SaTScan clusters of areas with a high prevalence of missing ENC among postpartum women in Ethiopia, EDHS 2016.**

| Most likely clusters | Enumeration areas (clusters) identified | Number of clusters | Population | No. of case | Coordinates / Radius | Relative risk | LLR | P-Value |
|---|---|---|---|---|---|---|---|---|
| 1st most likely cluster | 556, 394, 480, 187, 520, 318, 278, 208, 164, 358, 377, 85, 289, 286, 472, 138, 452, 7, 492, 422, 543, 92, 490, 198, 171, 95, 34, 146, 82, 497, 518, 123, 405, 562, 521, 588, 553, 26, 468, 316, 458, 601, 213,398, 319, 576, 313, 619, 529, 365, 600, 21, 245, 445, 232, 589, 12, 214, 372, 634, 251, 32, 182, 573, 476, 391, 574, 524, 239, 122, 308, 216, 578, 215, 116, 22, 408, 148 | 78 | 1033 | 746 | (5.330795 N, 41.837597 E)/427.38 km | 1.41 | 82.86 | <0.001 |
| 2nd most likely cluster | 138, 164, 85, 358, 146, 492, 92, 490, 543, 278, 171, 198, 95, 318, 77, 187, 497, 556, 520, 629, 521, 588, 553, 458, 480, 208, 214, 251, 573, 239, 269, 116, 22, 394, 378, 630, 568, 33, 277, 286, 527, 289 | 42 | 495 | 396 | (5.589269 N, 44.175032 E) / 443.03 km | 1.53 | 77.63 | <0.001 |
| 3rd most likely cluster | 4, 632, 75, 596, 440, 366, 178, 499, 205, 427, 334, 570, 348, 599, 544, 389, 368, 241, 55, 547, 191, 571, 344, 276, 332, 189, 254, 37, 249, 620, 488, 307, 135 | 33 | 400 | 313 | (11.845228 N, 41.915793 E) / 237.67 km | 1.49 | 53.33 | <0.001 |
| 4th most likely cluster | 403, 24, 429, 167, 456, 382, 120, 73, 516, 431, 375, 158, 169, 3, 512, 132, 474, 38, 206, 109, 361, 482, 531, 627, 229, 176, 163, 545, 350, 218, 10, 515, 292, 602, 498, 259, 541, 615, 494, 460, 548, 267, 510, 199, 386, 327, 415, 246, 533, 559, 640, 152, 354, 617, 312, 591, 616, 36, 150, 184, 401, 66, 279, 572, 628, 638, 423, 478, 183, 517 | 70 | 817 | 556 | (11.157729 N, 37.668699 E) / 206.74 km | 1.30 | 37.11 | <0.001 |
| 5th most likely cluster | 131, 16, 2, 250, 301, 323, 328, 356, 367, 379, 42, 48, 525, 530, 540, 561, 625, 641, 72, 9, 96, 618, 266, 309, 435, 536, 370, 507, 592, 260, 104 | 26 | 360 | 270 | (9.370004 N, 42.102751 E)/ 3855.29 km | 1.39 | 35.19 | <0.001 |
| 6th most likely cluster | 62, 411, 432, 586, 486, 447, 489, 227, 76, 142, 555, 280, 502, 294, 154, 118, 174, 234, 262, 177, 207, 577, 161, 399, 558, 331, 23, 306, 338, 477, 113, 272, 223, 41, 119 | 35 | 456 | 329 | (8.246355 N, 36.621500 E) / 154.70 km | 1.36 | 33.31 | <0.001 |
| 7th most likely cluster | 182, 574, 232, 32, 21, 316, 398, 600 | 8 | 122 | 106 | (5.973339 N, 38.182217 E) / 43.89 km | 1.62 | 30.63 | <0.001 |
| 8th most likely cluster | 476, 506, 412, 122, 333, 245, 372, 529, 71, 49, 491, 51, 230, 93, 564, 39, 484, 453, 336, 319, 441 | 20 | 267 | 194 | (8.888553 N, 40.744565 E) / 127.63 km | 1.36 | 20.12 | <0.001 |
| 9th most likely cluster | 134, 263, 192, 117 | 4 | 53 | 49 | (14.179123 N, 39.980749 E) / 28.29 km | 1.71 | 19.15 | 0.021 |
| 10th most likely cluster | 610 | 1 | 22 | 22 | (9.370004 N, 42.102751 E) / 0 km | 1.85 | 13.54 | 0.001 |
| 11th most likely cluster | 1, 566 | 2 | 32 | 30 | (9.505470 N, 42.438628 E) / 5.85 km | 1.74 | 12.55 | 0.0026 |
| 12th most likely cluster | 235, 585, 127 | 3 | 48 | 42 | (13.750028 N, 39.991260 E) / 15.05 km | 1.62 | 12.46 | 0.0028 |
| 13th most likely cluster | 562, 213, 619, 123 | 4 | 62 | 51 | (7.634301 N, 39.484475 E) / 50.25 km | 1.53 | 11.00 | 0.011 |
| 14th most likely cluster | 172 | 1 | 17 | 17 | (13.248133 N, 40.043685 E) / 0 km | 1.85 | 10.46 | 0.018 |
| 15th most likely cluster | 425, 80 | 2 | 28 | 26 | (13.351814 N, 38.353591 E) / 38.75 km | 1.72 | 10.36 | 0.021 |

**Table 5. Summary of OLS results for not receiving ENC in Ethiopia, EDHS 2016.**

| Variable | Coefficient | SE | t-Statistic | Probability | Robust SE | Robust t-statistics | Robust probability | VIF |
|---|---|---|---|---|---|---|---|---|
| Intercept | 0.303 | 0.043 | 2.41 | 0.016 | 0.047 | 2.19 | 0.028 | - - - |
| Women who gave birth at home | 0.674 | 0.032 | 16.57 | <0.001 | 0.042 | 12.82 | <0.001 | 1.19 |
| Women without formal education | 0.220 | 0.035 | 6.13 | <0.001 | 0.039 | 5.61 | <0.001 | 1.62 |
| Women without ANC | 0.501 | 0.033 | 14.93 | <0.001 | 0.036 | 13.87 | <0.001 | 1.76 |
| Women who never watch television | 0.171 | 0.040 | 4.24 | <0.001 | 0.048 | 3.51 | <0.001 | 1.94 |
| **OLS Diagnostics** | | | | | | | | |
| Number of Observations: | 622 | Akaike's Information Criterion (AICc) | | | | | | -349.68 |
| Multiple $R^2$ | 0.709 | Adjusted $R^2$ [d]: | | | | | | 0.718 |
| Joint F-Statistic | 377.40 | Prob(>F), (7,599) degrees of freedom | | | | | | <0.001 |
| Joint Wald Statistic | 2548.58 | Prob(>chi-square),(7) degrees of freedom | | | | | | <0.001 |
| Koenker (BP) Statistic | 41.62 | Prob(>chi-square), (7) degrees of freedom | | | | | | <0.001 |
| Jarque-Bera Statistic | 1.60 | Prob(>chi-square), (2) degrees of freedom | | | | | | 0.448 |

that the local model enhanced its ability to forecast hotspots of missing ENC. The highest range of $R^2$ was shown in central Addis Ababa, central and western Oromia, and the northern part of SNNPR regions. As compared to other regions, the model is less explained by the predictors in the southern part of the Benishangul Gumuz region (Fig 6).

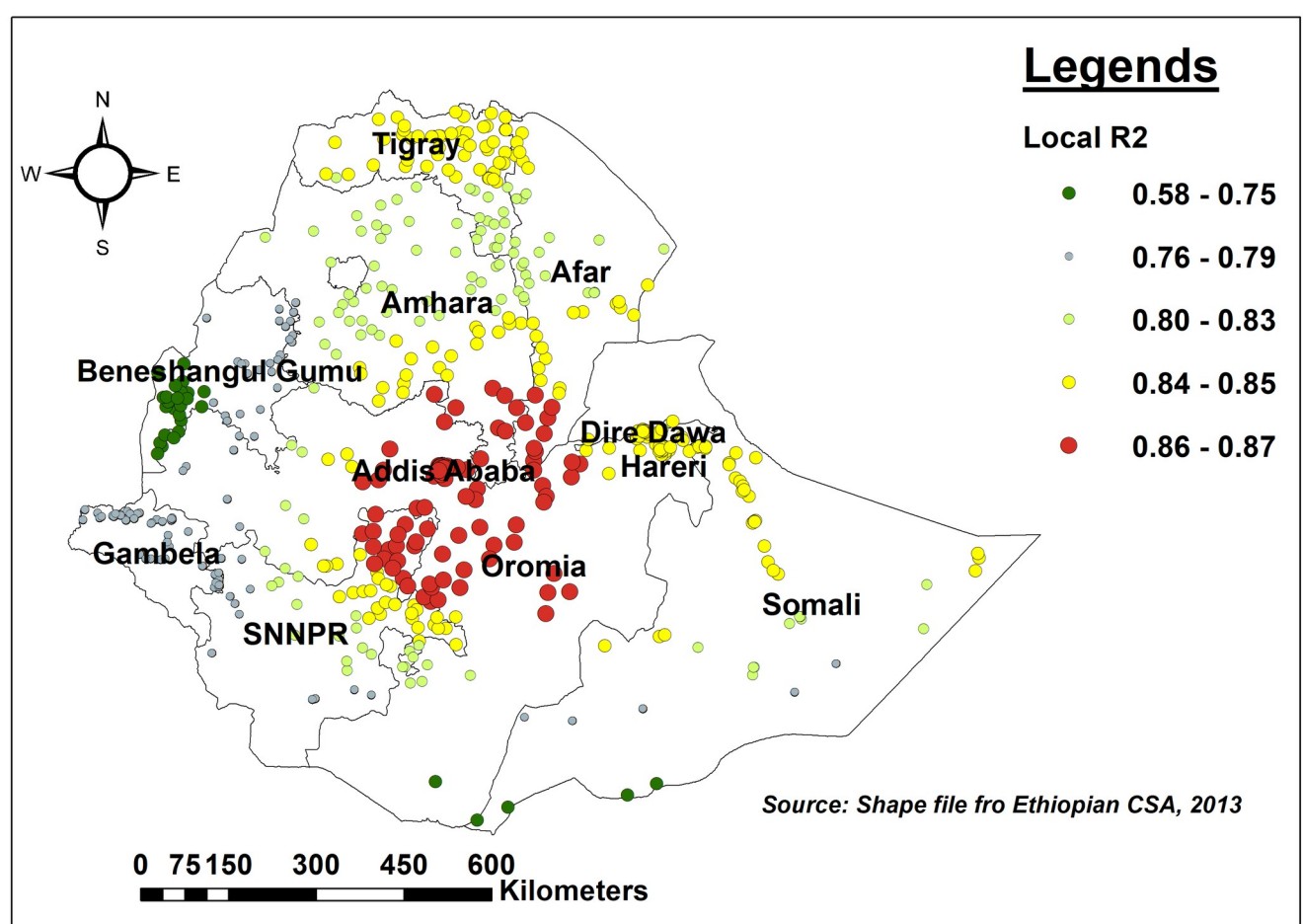

**Fig 6. The spatial mapping of local adjusted R-square of the GWR model in Ethiopia, 2016.**

## Proportion of women who gave birth at home

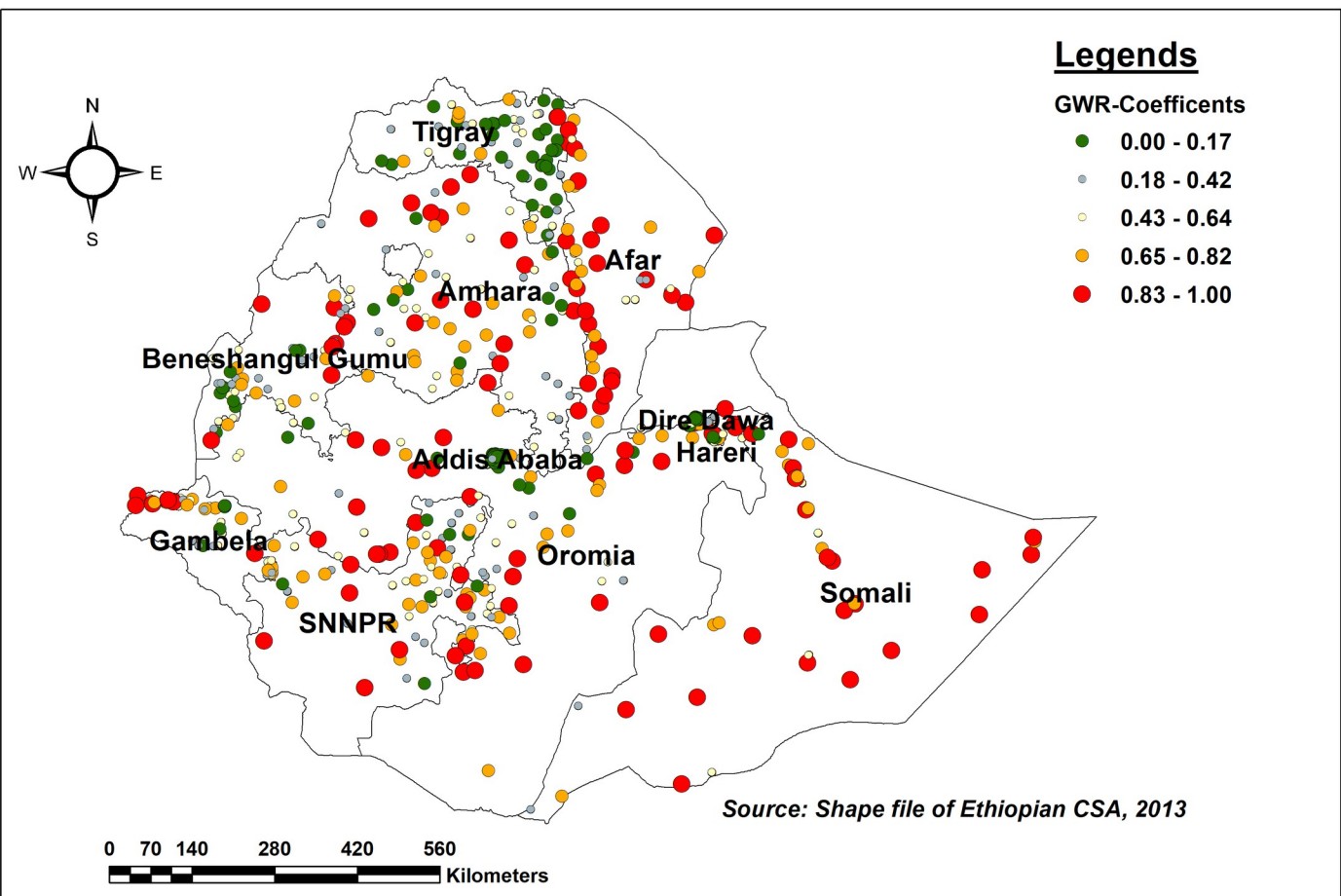

**Fig 7. GWR coefficients of the proportion of women who gave birth at home for predicting missing ENC messages in Ethiopia, EDHS 2016.**

As the proportion of women who gave birth at home increased, so did the proportion of missing ENC in most parts of Afar and Somali, in some parts of Oromia and Gambella, and the eastern border of Benishangul Gumuz. On the other hand, the positive and weaker relationship between home birth and missing ENC was observed in most parts of Tigray, eastern and central Amhara, and some parts of Addis Ababa (Fig 7).

This study also highlights the space-dependent relationship between missing ENC and education status. As the proportion of women with no formal education increases, the likelihood of missing ENC also increases in the entire Somali, west Afar, most parts of SNNPR, and the Northern part of Gambella regions (Fig 8).

Women who did not have ANC had a strong link with women who did not have ENC. As the proportion of women who did not attend ANC rose, so did the number of missing ENC in Afar, Somali, Northwest Gambella, south of the SNNPR, and northern and southern Amhara regions (Fig 9).

## Proportion of women with no formal education

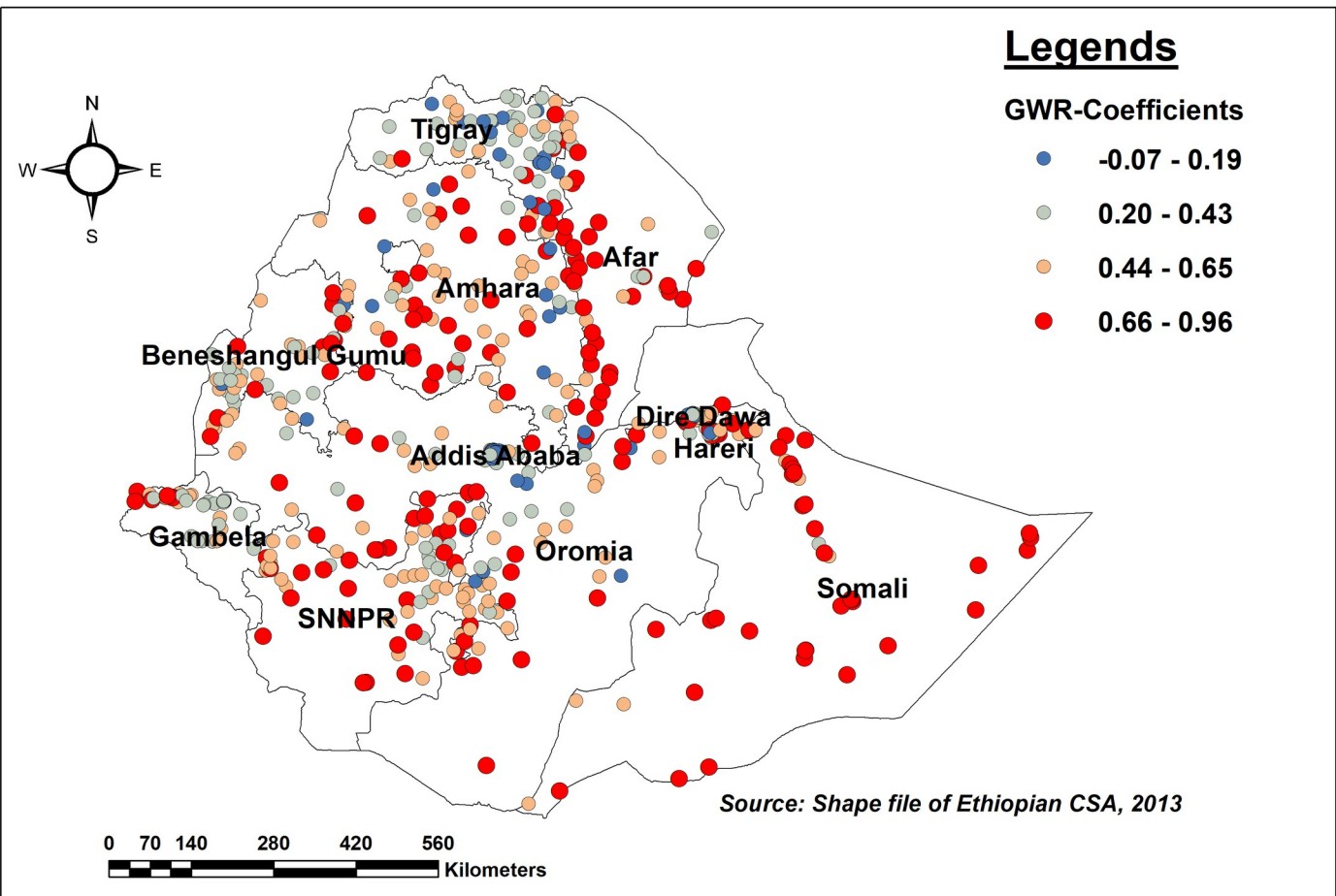

**Fig 8. GWR coefficients of the proportion of women without formal education for predicting missing ENC messages in Ethiopia, EDHS 2016.**

### Results of multilevel mixed effect negative binomial regression

**Random effect (measures of variation).**　In the null model, the value of ICC 0.261 which implies that 26.1% of the total variability in the receipt of ENC items was explained by the difference between clusters. Furthermore, variation at the individual and community levels accounted for 11.1% (ICC = 0.111, p<0.001) and 14.1% (ICC = 0.141, p<0.001), respectively, of the variation in the mean number of ENC items. Individual, and community-level factors together accounted for 51% of the mean variation seen in the null model (PCV = 73.2%). The values of AIC, BIC, and Deviance decreased as we progressed from model 1 (the empty model) to model 4 (the full model), indicating that the final model fitted throughout the study had adequate goodness of fit. Finally, the fourth model with the lowest deviance (17708.0) was chosen as the best model fit (Table 6).

**Fixed effects: Predictors of receipt of ENC items.**　In the multivariable multilevel negative binomial regression analysis, wealth index, educational status, frequency of ANC, residence, and region were identified as significant predictors of receiving ENC.

The incidence rate of getting ENC was lowered by 40% (aIRR = 0.60, 95%CI: 0.46, 0.79) and 33% (aIRR = 0.67, 95%CI: 0.51, 0.87) among women in the poorest and poorer wealth

**Fig 9. GWR coefficients of the proportion of women without ANC for predicting missing ENC messages in Ethiopia, EDHS 2016.**

quintiles, respectively than women in the richest wealth quintile. Women without formal education were 27% less likely to receive items of ENC than those who attend secondary or higher education (aIRR = 0.73, 95%CI: 0.60, 0.89). As compared to women who received 4 or more ANC visits, the chance of receiving ENC items fell by 48% among those who did not receive any ANC visits (aIRR = 0.52, 95%CI: 0.49, 0.71). Women who gave birth at home had a 73% lower likelihood of receiving ENC items (aIRR = 0.27, 95% CI: 0.17, 0.34) than women who gave birth in health facilities. Region and residence were the two community-level factors that were identified as significant predictors of ENC. Women who lived in the peripheral regions had 44% less chance of receiving ENC items (aIRR = 0.56, 95% CI: 0.47, 0.66) as compared to women who lived in metropolitan one. Similarly, women who resided in the rural part of the country were 61% less likely to receive ENC items as compared to their urban counterparts (aIRR = 0.39, 95% CI: 0.24, 0.57) (Table 6).

## Discussion

It is vital to investigate the spatial distribution and predictors of essential newborn care utilization at the national level to design a targeted intervention to reduce neonatal mortality and

**Table 6. Results of a multivariable mixed-effect negative binomial regression to identify the determinants of uptake of ENC items in Ethiopia, EDHS2016.**

| Variable categories | Model I (null model) | Model II (individual-level factors) | Model III (community-level factors) | Model-IV (full model) |
|---|---|---|---|---|
| | | IRR(95%CI) | IRR(95%CI) | IRR(95%CI) |
| **Current age** | | | | |
| 35–49 | | 1.01(0.83, 1.22) | | 0.95(0.78, 1.15) |
| 25–34 | | 1.07(0.93, 1.23) | | 1.05(0.91, 1.20) |
| 15–24 | | Ref. | | Ref. |
| **Wealth index** | | | | |
| Poorest | | 0.54(0.42, 0.69)** | | **0.60(0.46, 0.79)** |
| Poorer | | 0.59(0.46, 0.75)** | | **0.67(0.51, 0.87)** |
| Middle | | 0.83(0.69, 1.01) | | 0.79(0.62, 1.02) |
| Richer | | 0.85(0.71, 1.04) | | 0.81(0.62, 1.01) |
| Richest | | Ref. | | Ref. |
| **Educational status** | | | | |
| No education | | 0.71(0.59, 0.86)** | | **0.73(0.60, 0.89)** |
| Primary | | 0.85(0.72, 1.04) | | 0.87(0.73, 1.02) |
| Secondary and higher | | Ref. | | Ref. |
| **Head of household** | | | | |
| Male | | 0.86(0.76, 0.98)* | | 0.89(0.78, 1.01) |
| Female | | Ref. | | Ref. |
| **Parity** | | | | |
| Nulliparous | | 0.68(0.35, 1.35)* | | 0.69(0.34, 1.36) |
| Primiparous | | 1.15(0.95, 1.41) | | 1.12(0.92, 1.36) |
| Multiparous | | 0.95(0.81, 1.11) | | 0.92(0.79, 1.08) |
| Grand multiparous | | Ref. | | Ref. |
| **Frequency of ANC** | | | | |
| No visit | | 0.47(0.39, 0.56)** | | **0.52(0.49, 0.71)** |
| One visit | | 0.83(0.66, 1.04) | | 0.83(0.66, 1.04) |
| 2–3 visits | | 0.88(0.79, 1.08) | | 0.92(0.76, 1.15) |
| ≥4 visits | | Ref. | | Ref. |
| **Place of delivery** | | | | |
| Home | | 0.20(0.17, 0.24) | | **0.27(0.17, 0.34)** |
| Helath facility | | Ref. | | Ref. |
| **Listen to radio** | | | | |
| Not at all | | 0.95(0.79, 1.13) | | 0.93(0.77, 1.11) |
| Less than once a week | | 0.98(0.82, 1.19) | | 0.96(0.79, 1.16) |
| At least once a week | | Ref. | | Ref. |
| **Watching TV** | | | | |
| Not at all | | 0.89(0.74, 1.06) | | 0.98(0.82, 1.18) |
| Less than once a week | | 1.15(0.96, 1.38) | | 1.25(0.97, 1.52) |
| At least once a week | | Ref. | | Ref. |
| **Reading newspaper** | | | | |
| Not at all | | 1.15(0.93, 1.43) | | 1.14(0.91, 1.42) |
| Less than once a week | | 1.21(0.94, 1.62) | | 1.28(0.99, 1.60) |
| At least once a week | | Ref. | | Ref. |
| **Own mobile phone** | | | | |
| No | | 0.89(0.74, 1.05) | | 0.90(0.75, 1.07) |
| Yes | | Ref. | | Ref. |
| **Covered by Health Insurance** | | | | |

*(Continued)*

**Table 6.** (Continued)

| Variable categories | Model I (null model) | Model II (individual-level factors) | Model III (community-level factors) | Model-IV (full model) |
|---|---|---|---|---|
| No | | 0.86(0.72, 1.09) | | 0.95(0.77, 1.16) |
| Yes | | Ref. | | Ref. |
| **Autonomy in decision-making** | | | | |
| Non-autonomous | | 1.15(0.83, 1.05) | | 0.94(0.83, 1.06) |
| Autonomous | | Ref. | | Ref. |
| **Distance to seek healthcare** | | | | |
| Big problem | | 0.84(0.75, 0.95)** | | 0.89(0.78, 1.02) |
| Not a big problem | | Ref. | | Ref. |
| **Ease of access to money** | | | | |
| Big problem | | 0.97(0.85, 1.10) | | 0.96(0.85, 1.10) |
| Not a big problem | | Ref. | | Ref. |
| **Regions** | | | | |
| Peripheral | | | 0.44(0.36, 0.53)** | **0.56(0.47, 0.66)**** |
| Major central regions | | | 0.65(0.56, 0.76)** | 0.83(0.71, 0.98)** |
| Metropolitans | | | Ref. | Ref. |
| **Residence** | | | | |
| Rural | | | 0.27(0.23, 0.32)** | **0.39(0.24, 0.57)**** |
| Urban | | | Ref. | Ref. |
| **Random effects** | | | | |
| Variance | 1.16 | 0.41 | 0.54 | 0.31 |
| ICC | 0.261 | 0.111 | 0.141 | 0.086 |
| AIC | 18847.6 | 17890.3 | 18544.6 | 16788.11 |
| BIC | 18868.2 | 18089.8 | 18640.0 | 17063.3 |
| MOR | 2.80 | 1.85 | 2.01 | 1.71 |
| PCV | Ref. | 64.6% | 53.4% | 73.2% |
| **Model fitness** | | | | |
| Log-likelihood | -9420.8 | -8177.2 | -9319.5 | -8854.0 |
| Deviance | 18841.6 | 18354.4 | 18639.0 | 17708.0 |

**Key:** Ref.: Reference category; aIRR = Adjusted Incidence Rate Ratio,

* statistically significant at p-value <0.05,

** statistically significant at p-value <0.001

increase satisfaction and utilization of maternal and neonatal health services by women [26]. In this study, the prevalence of missing ENC was found to be 61.6% (95% CI: 60.5, 62.7) which is higher than a systematic review and meta-analysis conducted in Ethiopia (51.2%) [17], and similar studies conducted in Nepal (29.3%) [13], India (32.3%) [14], Rwanda(35.5%) [15], and Tanzania(31%) [16]. The disparity could be attributed to differences in socioeconomic and sociocultural characteristics, availability and accessibility of health service infrastructure, and maternal health service coverage, between countries.

In addition, spatial global Moran's analysis found that the percentage of women who didn't receive ENC varied geographically. As per hot spot analysis, statistically significant hotspot areas for missing ENC were Somali, central and southwest Afar, south Amhara, and the eastern border of SNNPR regions. Previous studies revealed that maternal and neonatal health service uptake was low in those regions [49–53]. This clustering might be due to a variety of factors. To begin, as compared to other regions, those two regions, particularly Afar and

Somali, are known to have a shortage of healthcare facilities, limited healthcare providers, and a lack of medical supplies and equipment to provide ENC services [54, 55]. In addition, they are located in remote and arid areas with inadequate road infrastructure and transportation services, making it difficult for both women and health care providers to access and deliver immediate postnatal care services, potentially leading to high coverage of missing ENC [56]. Furthermore, because the majority of the people in these regions live pastoral or partly nomadic lives, it may be difficult for postpartum women to access adequate ENC services because their places of residence change over time.

In multilevel mixed effect negative binomial and Geographically weighted regression analyses, having no formal education, giving birth at home, and lack of ANC were identified as significant predictors of not receiving ENC in Ethiopia. In addition, living in the poorest wealth quintile, being a rural resident, and living in a peripheral region were also identified as significant determinants of the uptake of ENC items in a multilevel regression model.

Not receiving ANC for the last pregnancy was found to be strongly associated with missing ENC items. This was supported by studies conducted in Bangladesh [57], Nigeria [58], Ghana [59], Uganda [60], East Africa [61] and Ethiopia [62]. In addition, studies conducted in Asia revealed that information on newborn care practices provided to pregnant women and their families during ANC visits resulted in enhanced ENC practices such as complete cord care, complete thermal care, and breastfeeding initiation [63]. In addition, ANC links pregnant women with skilled healthcare providers who can assist with safe deliveries and immediate newborn care [64]. Lack of ANC visits could result in missed opportunities for newborn care education and information, reduced access to skilled birth attendants, and inadequate birth preparedness and complication readiness, all of which can contribute to a higher likelihood of missing ENC, which is critical for ensuring the health and survival of newborns [58].

Home birth was also identified as a significant predictor of missing ENC packages. This was in tandem with the findings of studies conducted elsewhere [65–67]. This might be due to a variety of reasons. First, giving birth at home potentially can lead to difficulty in accessing skilled healthcare providers in the immediate postpartum period, and this results in missing ENC [65]. As compared to births at health facilities, those at home have a lower or no likelihood of receiving postnatal care, education, and counselling, and being closely monitored by healthcare providers resulted in missing vital ENC components. Thus, efforts should focus on increasing access to skilled birth attendants through improving healthcare and transportation infrastructure, educating communities about the importance of facility deliveries, and addressing cultural and socioeconomic factors that hinder delivery choices [68, 69].

The current study revealed that the risk of receiving ENC items was lower among women who never attended formal education. This was supported by studies conducted in LMICs [70], Pakistan, Ghana [59], and Ethiopia [71]. This could be due to, women without formal education may have limited access to information about ENC and its importance, which can lead to suboptimal care. Furthermore, a lack of formal education may result in a lower socioeconomic status and accompanying financial constraints, as well as a lack of autonomy in decision-making, all of which can be linked to limiting access to the resources needed and healthcare services for ENC. As a result, improving access to formal education is essential for addressing those challenges, particularly in areas where formal education is limited.

Similarly, living in a household with the poorest wealth quintile was revealed to be a statistically significant predictor of not receiving ENC. This was supported by studies conducted in India [72], Nepal [73], and Ethiopia [74]. Those women in the poorest wealth quintile are more likely to experience financial barriers in accessing maternal healthcare, especially ANC, skilled delivery, and PNC services all of which are vital entry points to get ENC services.

Furthermore, poverty and financial constraints can put a strain on women and families by limiting access to information and the ability to make health-care decisions [75].

The study has the following strengths. First, the findings were based on an analysis of nationally representative data, making the findings more generalizable. This study is also enriched by the findings of Hot Spot, geographically weighted regression, and multilevel mixed-effect negative binomial regression analysis. As a result, the findings can assist government and program planners in designing geographical, individual, and community-focused public health interventions based on the identified predictors for tackling barriers to ENC. On the other hand, the finding also should be interpreted in light of limitations. To begin, to safeguard the confidentiality of respondents or the community, the geographical coordinates of clusters were displaced by up to 2km in urban areas, 5km for most rural clusters, and 10km for 1% of rural clusters; this may alter estimated cluster effects in the spatial regression. Second, because of the cross-sectional nature of EDHS data, it is hard to infer a temporal/causal relationship between the variables. Furthermore, as the data were based on the retrospective interviews of women who responded to receipt of PNC for their last birth, the findings may be subject to recall bias. Finally, the data were based on self-report and this tends to face recall- and social desirability bias which may underestimate or overestimate the true association.

## Conclusion

The level of missing ENC during immediate PPP was found to be high in Ethiopia with a significant spatial variation across regions. Statistically significant hotspot areas for not receiving ENC were northern Somali, central and southwest Afar, the southern part of Amhara, and the eastern border of SNNPR regions. Living in the poorest wealth quintile, being a rural resident, having no formal education, giving birth at home, and lack of ANC were identified as significant predictors of not receiving ENC. Hence, the government and policymakers should devise strategies for hotspot areas to improve women's economic capabilities, access to education, and health-seeking behaviours for prenatal care and skilled delivery services in order to improve ENC uptake.

## Acknowledgments

We are grateful to ICF macro (Calverton, USA) for providing the 2016 DHS data of Ethiopia.

## Author Contributions

**Conceptualization:** Aklilu Habte Hailegebireal.

**Data curation:** Aklilu Habte Hailegebireal.

**Formal analysis:** Aklilu Habte Hailegebireal, Aiggan Tamene Kitila.

**Methodology:** Aklilu Habte Hailegebireal, Aiggan Tamene Kitila.

**Software:** Aklilu Habte Hailegebireal.

**Supervision:** Aklilu Habte Hailegebireal.

**Visualization:** Aklilu Habte Hailegebireal.

**Writing – original draft:** Aklilu Habte Hailegebireal, Aiggan Tamene Kitila.

**Writing – review & editing:** Aklilu Habte Hailegebireal, Aiggan Tamene Kitila.

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
