## [Decision Letter · Decision Letter 0]

24 Apr 2024

PONE-D-23-29356Spatial patterns and predictors of missing essential newborn care items during the immediate postpartum period in Ethiopia: Spatial and multilevel count analyses of 2016 Demographic Health SurveyPLOS ONE

Dear Dr. Habte,

Thank you for submitting your manuscript to PLOS ONE. After careful consideration, we feel that it has merit but does not fully meet PLOS ONE’s publication criteria as it currently stands. Therefore, we invite you to submit a revised version of the manuscript that addresses the points raised during the review process.

We look forward to receiving your revised manuscript.

Kind regards,

Amanuel Abajobir, PhD

Academic Editor

PLOS ONE

2. ote from Emily Chenette, Editor in Chief of PLOS ONE, and Iain Hrynaszkiewicz, Director of Open Research Solutions at PLOS: Did you know that depositing data in a repository is associated with up to a 25% citation advantage (https://doi.org/10.1371/journal.pone.0230416)? If you’ve not already done so, consider depositing your raw data in a repository to ensure your work is read, appreciated and cited by the largest possible audience. You’ll also earn an Accessible Data icon on your published paper if you deposit your data in any participating repository (https://plos.org/open-science/open-data/#accessible-data).

3. In the online submission form, you indicated that [The data for this study were obtained from the DHS program with a reasonable request. Thus, the one who needs the data supporting the findings of this study can get it in anonymized form from the DHS website at https://www.dhsprogram.com upon reasonable request in the same manner as the authors did.]. 

4. We note that Figure(s) [1,4,5,6,7,8 and 9,] in your submission contain [map/satellite] images which may be copyrighted. All PLOS content is published under the Creative Commons Attribution License (CC BY 4.0), which means that the manuscript, images, and Supporting Information files will be freely available online, and any third party is permitted to access, download, copy, distribute, and use these materials in any way, even commercially, with proper attribution. For these reasons, we cannot publish previously copyrighted maps or satellite images created using proprietary data, such as Google software (Google Maps, Street View, and Earth). For more information, see our copyright guidelines: http://journals.plos.org/plosone/s/licenses-and-copyright.

1. You may seek permission from the original copyright holder of Figure(s) [1,4,5,6,7,8 and 9,] to publish the content specifically under the CC BY 4.0 license.  

Additional Editor Comments:

Overall, the manuscript provides a comprehensive overview of the neonatal care landscape in Ethiopia, particularly focusing on the postnatal period and essential newborn care (ENC) services. Here's a breakdown of the review focusing on the quality of English, writing, and content:

Quality of English and writing:

There are a few instances where sentences could be refined for clarity and flow. For instance, in the Methods and Materials section, sentences like "Respondents were selected using a stratified two-stage cluster sampling technique, having each region divided into urban and rural areas" could be rephrased for smoother reading, such as "Respondents were selected through a stratified two-stage cluster sampling technique, where each region was divided into urban and rural areas." Such minor adjustments can enhance readability without altering the technical content.

Additionally, there are some instances of lengthy sentences and complex structures that could be simplified for better comprehension, especially for readers not familiar with the subject matter. Breaking down complex sentences into smaller, digestible parts can improve readability and understanding.

Content:

Introduction: The introduction provides a comprehensive overview of the significance of the neonatal period and the challenges faced, especially in low-resource settings like Ethiopia. It effectively sets the stage for the study by highlighting the gap in essential newborn care utilization. However, it should be clear and concise in setting the stage!

Methods and Materials: Some parts could benefit from concise explanations to avoid overwhelming the reader with technical details. For example, the explanation of spatial autocorrelation and spatial interpolation could be simplified for clarity.

Results/Discussion: See above for the quality of writing.

Reviewers' comments:

Reviewer's Responses to Questions

**Comments to the Author**

1. Is the manuscript technically sound, and do the data support the conclusions?

Reviewer #1: Partly

2. Has the statistical analysis been performed appropriately and rigorously? 

Reviewer #1: Yes

3. Have the authors made all data underlying the findings in their manuscript fully available?

Reviewer #1: No

4. Is the manuscript presented in an intelligible fashion and written in standard English?

Reviewer #1: No

5. Review Comments to the Author

Reviewer #1: Title: Spatial patterns and predictors of missing essential new born care items during the immediate postpartum period in Ethiopia: Spatial and multilevel count analyses of 2016

Demographic Health Survey is a well-written manuscript. Some comments to consider before publication are listed below.

General comments

- This 2016 EDHS data is too old, we do have MEDHS of 2019 data, is it justifiable that you can still go back to 2016 while new data available? considering trends could enable you to use all data…..probably

- Your abstract method section does not tell us which ANOVA you used

- Why SPSS 16, while you can do 25 or more what is the secret?

- The abstract does not list what is the essential newborn care or defining variables

- Abstract result focuses only on spatial results like there are no other variables

- Include the success number with percent in abstract and result sections …..(()60%)

- The level ENC is lower in your abstract compared to nothing……. Please avoid partial statements as this is a common problem throughout. For example, the government and policymakers should tailor strategies to hotspot areas by implementing activities and interventions…. What strategies…??? What implementation??

- Just avoid general statements and non-practical recommendations or conclusions, this makes the study nonsense.

Introduction

- However, the drop in NNM has been more static, falling from 58 deaths per 1000 live births in 2000 to 29 deaths per 1000 births in 2016. This statement is not clear. I do know how something static could have this much variation. Please, follow the academic way of writing, your writing lacks details and makes partial statements and it is not to the point of focus. You need to go through the whole document and make clear sense of those limited ideas.

- Is PPP just 42 days? Exactly what information is necessary for a mother during postpartum?

- intrapartum and postpartum care remained mostly responsible for a steady reduction in neonatal death in Ethiopia……what are those intrapartum and postpartum care missing in Ethiopia?

- Your finding indicates 60% of ENC is missing, but your literature does not have any information related to this excepting million global deaths. You need to reconstruct your introduction according to your main objective.

- Based on this evidence the gap for which you conducting this study is clearly identified and you need to deal more.

Methods

Your methods look nice but consider the following

- Have clear data preparation and how you handled missing or if other difficulties exist as this data has many gaps

- Provide a clear procedure for your analysis and assumptions for low scholars' understanding. You have that for spatial analysis but not for others

- How do you apply OLS with dichotomous data?

Results

- You have very large result presentations, you may need to make a summary of results at the end

- Somalia, Afar, and Gambella are usually underperforming in most studies what is different about them?

Discussion

- Result comparison could make more sense if you can make a more focus comparison with similar setups and reasonably compare how that is common is most developing countries. Something is missing in your initial paragraph – focus

Conclusion

- I already commented on how to make the conclusion more practical earlier, thus, apply here.

-

6. PLOS authors have the option to publish the peer review history of their article (what does this mean?). If published, this will include your full peer review and any attached files.

Reviewer #1: No

---

## [Author Response · Author response to Decision Letter 0]

3 May 2024

A point-by-point response to editor and reviewers

Authors’ Response to Academic Editor

Dear: Amanuel Abajobir, PhD, Academic Editor, Plos One

We thank you for a thorough reading and constructive comments and suggestions on our manuscript and for the opportunity to revise and resubmit. We are pleased to submit the revised version of the manuscript titled “Geographical variation and predictors of missing essential newborn care during the immediate postpartum period in Ethiopia: Spatial and multilevel count analyses” for your consideration in the special collection of Plos One. The comments of the editors and the reviewers were highly insightful and enabled us to greatly improve the quality of our manuscript. In this revised manuscript we made substantial changes to address your concerns in a point-by-point response. We appreciate your time and look forward to your response and we are very keen to incorporate further comments, if any, for the betterment of the final manuscript.

On the following pages, you will find our responses to the comments and suggestions raised by the esteemed editor and reviewer. 

Sincerely, 

Aklilu Habte (MPH)(corresponding author)

aklilihabte57@gmail.com

Response to Journal requirements

Response: we already prepared the manuscript as per the journal requirement and again we rechecked the compliance towards it during the submission of our revised manuscript.

2. In the online submission form, you indicated that [The data for this study were obtained from the DHS program with a reasonable request. Thus, the one who needs the data supporting the findings of this study can get it in anonymized form from the DHS website at https://www.dhsprogram.com upon reasonable request in the same manner as the authors did.]. 

Response: The DHS data policy doesn’t allow sharing of the dataset with third parties for other than initially registered purposes. Thus, we suggested the readers access the data on the above-mentioned link based on the reasonable request.

3. We note that Figure(s) [1,4,5,6,7,8 and 9,] in your submission contain [map/satellite] images which may be copyrighted. All PLOS content is published under the Creative Commons Attribution License (CC BY 4.0), which means that the manuscript, images, and Supporting Information files will be freely available online, and any third party is permitted to access, download, copy, distribute, and use these materials in any way, even commercially, with proper attribution. For these reasons, we cannot publish previously copyrighted maps or satellite images created using proprietary data, such as Google software (Google Maps, Street View, and Earth). For more information, see our copyright guidelines: http://journals.plos.org/plosone/s/licenses-and-copyright.We require you to either (1) present written permission from the copyright holder to publish these figures specifically under the CC BY 4.0 license, or (2) remove the figures from your submission:

 1. You may seek permission from the original copyright holder of Figure(s) [1,4,5,6,7,8 and 9,] to publish the content specifically under the CC BY 4.0 license. 

Response: We appreciate your concern to assure the ethical issues. However, all the aforementioned figures (1,4,5,6,7,8 and 9) in our manuscript are not copyrighted rather they are the result of spatial analysis that we have run in ArcGIS and SaTScan software. The GPS and DHS data that contain Shapefile and other relevant variables were obtained from the DHS office by explaining the objective of the study through online requests. Then, in order to get those figures, we import the relevant data extracted from the 2016 Ethiopian Demographic Health Survey reports and the shapefile of Ethiopia obtained from the 2016 Ethiopian Central Statistical Agency (CSA).To indicate this, we already cited the source of the shapefile alongside each figure. The shape file that we used to construct the figures can be accessed by one of the following links:

1. https://data.humdata.org/dataset/cb58fa1f-687d-4cac-81a7-655ab1efb2d0

2. https://gadm.org/download_country.html

Therefore, the maps presented in our study are not copyrighted rather they were the outputs of our spatial analysis results which are the result of those Shapefiles and projected CVS files in ArcGIS. This is the actual procedure that we employed in our present and earlier studies, as well as other Ethiopian researchers. Again we assure you that the figures presented in our study are not copyrighted but rather our spatial analysis results.

Response to Additional Editor Comments: 

General Comment: Overall, the manuscript provides a comprehensive overview of the neonatal care landscape in Ethiopia, particularly focusing on the postnatal period and essential newborn care (ENC) services. Here's a breakdown of the review focusing on the quality of English, writing, and content:

Response: Thank you for your positive and constructive comments and suggestions we got all of them crucial in the improvement of the manuscript. Accordingly, we tried to respond to your possible suggestions as follows:

Comment 1: There are a few instances where sentences could be refined for clarity and flow. For instance, in the Methods and Materials section, sentences like "Respondents were selected using a stratified two-stage cluster sampling technique, having each region divided into urban and rural areas" could be rephrased for smoother reading, such as "Respondents were selected through a stratified two-stage cluster sampling technique, where each region was divided into urban and rural areas." Such minor adjustments can enhance readability without altering the technical content. Additionally, there are some instances of lengthy sentences and complex structures that could be simplified for better comprehension, especially for readers not familiar with the subject matter. Breaking down complex sentences into smaller, digestible parts can improve readability and understanding.

Response: Thank you for your insightful suggestions. We tried to correct some vague and complex sentences and highlighted them throughout the "Revised Manuscript with Track Changes"

Comment 2: Methods and Materials: Some parts could benefit from concise explanations to avoid overwhelming the reader with technical details. For example, the explanation of spatial autocorrelation and spatial interpolation could be simplified for clarity.

Response: Thank you for your suggestion. Initially, we did this intending to make the statements more clear to readers. Now, we entirely concur with your point of view because detailed explanations can be confusing to readers, thus we attempted to simplify certain difficult remarks.

Thank you for your constructive comments and suggestions, which we got as valuable input in the improvement of the manuscript. We received all of them as a valuable contribution to our ongoing work. In the following section, we tried to respond to all the possible comments and suggestions from reviewer #1 

END________________________________________

 THANK YOU!!!

Authors’ Response to Reviewer#1 

General comment: Demographic Health Survey is a well-written manuscript. Some comments to consider before publication are listed below. 

Response: Dear Reviewer 1, thank you very much for taking the time to review our work and for your positive feedback. We received your thoughtful, and generous review, along with helpful feedback and suggestions, as a valuable contribution to our ongoing work. We have tried to address all the possible comments and suggestions raised by you in the following session.

Comment 1: Your abstract method section does not tell us which ANOVA you used

Response: thank you for your meticulous review. It was to mean a One-way ANOV and we have corrected and highlighted it in the ‘Abstract’ section of the "Revised Manuscript with Track Changes" Page 2, Line 50

Comment 2: Why SPSS 16, while you can do 25 or more what is the secret? 

Response: It was not SPSS v16 it was STATA version 16 which was the most updated version at hand during the time of our analysis. 

Comment 3: The abstract does not list what is the essential newborn care or defining variables

Response: Thank you for your thoughtful inquiry. As the abstract should be with a limited amount of words, initially we tried to make it short and precise. Now, as per your suggestion, we incorporated the items and highlighted them in the ‘Abstract’ section of the "Revised Manuscript with Track Changes" Page 2, Line 46-49

Comment 4: Abstract result focuses only on spatial results like there are no other variables

Response: We tried to incorporate the results of both spatial and multilevel analysis in a more short and precise way. For your clarity, the results other than spatial analysis were highlighted in the ‘Abstract’ section of the "Revised Manuscript with Track Changes" Page 2, Lines 58-63. Comment 5: Include the success number with percent in abstract and result sections …..(()60%)

Response: We have corrected and highlighted it in the ‘Abstract’ section of the "Revised Manuscript with Track Changes" Page 2, Line 56.

Comment 6: The level ENC is lower in your abstract compared to nothing……. Please avoid partial statements as this is a common problem throughout. For example, the government and policymakers should tailor strategies to hotspot areas by implementing activities and interventions…. What strategies…??? What implementation?? Just avoid general statements and non-practical recommendations or conclusions, this makes the study nonsense.

Response: Thank you for your inquiry and constructive suggestions. From the start, we concluded that Ethiopia had a higher level of missing ENC. The conclusion was reached after comparing and contrasting the current statistics with previous large-scale studies undertaken in Ethiopia and other countries. For that matter, we kindly ask you to look into the ‘Discussion’ section of the "Revised Manuscript with Track Changes" Page 21, Lines 454-457. Regarding our recommendation, it lacks specificity and thus we made an amendment and highlighted it in the ‘Abstract’ section of the "Revised Manuscript with Track Changes" Page 3, Lines 66-68 

Comment 7: However, the drop in NNM has been more static, falling from 58 deaths per 1000 live births in 2000 to 29 deaths per 1000 births in 2016. This statement is not clear. I do know how something static could have this much variation. Please, follow the academic way of writing, your writing lacks details and makes partial statements and it is not to the point of focus. You need to go through the whole document and make clear sense of those limited ideas.

Response: It was to mean the figure was not satisfactory as per the government plan and to show it is vital to work on newborns. We have made amendments to the statement and highlighted it in the ‘Introduction’ section of the "Revised Manuscript with Track Changes" Page 3, Lines 81-82. We gave due emphasis to your comments and suggestions throughout the revised version of the manuscript and we have highlighted them.

Comment 8: Is PPP just 42 days? Exactly what information is necessary for a mother during postpartum?

Response: Yes, the postpartum period covers from the time of delivery to 42 days postpartum. However, our study mainly focuses on the contents of care provided during the immediate postpartum period (i.e. within 2 days following birth) which is the focus of DHS data. In addition, other preventive and curative services could be provided during PPP but they were not our current study’s interest.

Comment 9: intrapartum and postpartum care remained mostly responsible for a steady reduction in neonatal death in Ethiopia……what are those intrapartum and postpartum care missing in Ethiopia?

Response: Thank you for asking. The main intrapartum care was skilled delivery service and the postpartum one was essential newborn care services. We made slight amendments to the statement and highlighted it in the ‘Introduction’ section of the "Revised Manuscript with Track Changes" Page 3, Lines 104-106.

Comment 10: Your finding indicates 60% of ENC is missing, but your literature does not have any information related to this excepting million global deaths. You need to reconstruct your introduction according to your main objective.

Response: Thank you for your suggestion. Accordingly, we added some figures regarding the level of missing ENC from previously conducted large-scale studies. That statement was highlighted in the ‘Introduction’ section of the "Revised Manuscript with Track Changes" Page 3, Lines 101-104.

Comment 10: Based on this evidence the gap for which you conducting this study is clearly identified and you need to deal with more. 

Response: thank you for your suggestion to strengthen the real gaps that were addressed by this study. Accordingly, we highlighted the gaps that were addressed by the current study in the last paragraph of the ‘Introduction’ section of the "Revised Manuscript with Track Changes" Page 4.

Comment 11: Have clear data preparation and how you handled missing or if other difficulties exist as this data has many gaps

Response: we have removed those respondents without any information on the uptake of PNC within 42 days. We have mentioned this in the ‘Population of the study’ section of the "Revised Manuscript with Track Changes" Page 5, Lines 138-142. Beyond that, we didn’t face any difficulty with missing values on other important covariates. 

Comment 12: Provide a clear procedure for your analysis and assumptions for low scholars' understanding. You have that for spatial analysis but not for others

Response: Thank you for your insightful suggestion. We have mentioned the details of assumptions that we followed while running a generalized linear model (a multilevel negative binomial regression) in the ‘Data management and statistical analysis’ section of the "Revised Manuscript with Track Changes" Page 10, Lines 257-264. 

Comment 13: How do you apply OLS with dichotomous data?

Response: The ordinary least square(OLS) that we applied in the current study was from the spatial regression dimension, not from the perspective of ordinary linear regression. we kindly ask you to look into the details of the spatial regression section. 

Comment 14: You have very large result presentations, you may need to make a summary of results at the end

Response: thank you for your suggestion. However, the journal requirement doesn’t allow us to add the summary part to the result section. The only section to summarize our findings was the ‘Abstract’ section and we tried to summarize all the main findings here. By emphasizing your remarks, we attempted to make the result section more concise and precise by trimming some of the lengthy phrases.

Comment 15: Somalia, Afar, and Gambella are usual

---

## [Editor Report · Decision Letter 1]

22 May 2024

PONE-D-23-29356R1Geographical variation and predictors of missing essential newborn care during the immediate postpartum period in Ethiopia: Spatial and multilevel count analysesPLOS ONE

Dear Dr. Habte,

Thank you for submitting your manuscript to PLOS ONE. After careful consideration, we feel that it has merit but does not fully meet PLOS ONE’s publication criteria as it currently stands. Therefore, we invite you to submit a revised version of the manuscript that addresses the points raised during the review process.

We look forward to receiving your revised manuscript.

Kind regards,

Amanuel Abajobir, PhD

Academic Editor

PLOS ONE

Journal Requirements:

**Additional Editor Comments:**

**Although the revised version has addressed most of the comments, significant language edits are still required to meet the standards of scientific writing before it can be formally accepted for publication.**

---

## [Author Response · Author response to Decision Letter 1]

23 May 2024

A point-by-point response to editor and reviewers

Authors’ Response to Academic Editor

Dear: Amanuel Abajobir, PhD, Academic Editor, Plos One

We thank you for a thorough reading and constructive comments and suggestions on our manuscript and for the opportunity to revise and resubmit. We are pleased to submit the revised version of the manuscript titled “Geographical variation and predictors of missing essential newborn care during the immediate postpartum period in Ethiopia: Spatial and multilevel count analyses” for your consideration in the special collection of Plos One. The comments of the editors and the reviewers were highly insightful and enabled us to greatly improve the quality of our manuscript. In this revised manuscript we made substantial changes to address your concerns in a point-by-point response. We appreciate your time and look forward to your response and we are very keen to incorporate further comments, if any, for the betterment of the final manuscript.

Sincerely, 

Aklilu Habte (MPH)(corresponding author)

aklilihabte57@gmail.com

Response to Journal requirements

Response: Thank you for your reminder to look into the appropriateness of the references that we used for this work. Accordingly, we got all the references were relevant and none of them were retracted. 

Response to Additional Editor Comments: 

Comment 1: Although the revised version has addressed most of the comments, significant language edits are still required to meet the standards of scientific writing before it can be formally accepted for publication.

Response: Thank you for acknowledging that we addressed most of the comments raised by the reviewers and editor. As per your suggestion, we tried to give due emphasis to the grammatical and typological issues throughout the manuscript and we tried to correct and highlight them throughout the "Revised Manuscript with Track Changes"

Comment 2: While revising your submission, please upload your figure files to the Preflight Analysis and Conversion Engine (PACE) digital diagnostic tool, https://pacev2.apexcovantage.com/. PACE helps ensure that figures meet PLOS requirements. To use PACE, you must first register as a user. Registration is free. Then, login and navigate to the UPLOAD tab, where you will find detailed instructions on how to use the tool. If you encounter any issues or have any questions when using PACE, please email PLOS at figures@plos.org Please note that Supporting Information files do not need this step.

Response: Thank you for your suggestion. However, we initially submitted the figures as per the journal’s requirement and we want to assure you that they were PACE corrected figures.

Thank you for your constructive comments and suggestions, which we got as valuable input in the improvement of the manuscript. We received all of them as a valuable contribution to our ongoing work. 

END________________________________________

 THANK YOU!!!

---

## [Decision Letter · Decision Letter 2]

3 Jul 2024

PONE-D-23-29356R2Geographical variation and predictors of missing essential newborn care during the immediate postpartum period in Ethiopia: Spatial and multilevel count analysesPLOS ONE

Dear Dr. Habte,

Thank you for submitting your manuscript to PLOS ONE. After careful consideration, we feel that it has merit but does not fully meet PLOS ONE’s publication criteria as it currently stands. Therefore, we invite you to submit a revised version of the manuscript that addresses the points raised during the review process.

We look forward to receiving your revised manuscript.

Kind regards,

Amanuel Abajobir, PhD

Academic Editor

PLOS ONE

Additional Editor Comments:

Please remove any reference to any 'ethnicity' in the 'Classification' of the submission system.

Reviewers' comments:

Reviewer's Responses to Questions

**Comments to the Author**

1. If the authors have adequately addressed your comments raised in a previous round of review and you feel that this manuscript is now acceptable for publication, you may indicate that here to bypass the “Comments to the Author” section, enter your conflict of interest statement in the “Confidential to Editor” section, and submit your "Accept" recommendation.

Reviewer #2: (No Response)

2. Is the manuscript technically sound, and do the data support the conclusions?

Reviewer #2: Yes

3. Has the statistical analysis been performed appropriately and rigorously? 

Reviewer #2: Yes

4. Have the authors made all data underlying the findings in their manuscript fully available?

Reviewer #2: Yes

5. Is the manuscript presented in an intelligible fashion and written in standard English?

Reviewer #2: Yes

6. Review Comments to the Author

Reviewer #2: The manuscript requires major editorial editing throughout. The cited references, abbreviations etc need to be spaced out from the words and figures consistently across the entire manuscript, including the tables. Please refer to a PLOS journal article for an example.

The analysis conducted is comprehensive, but the manuscript could be further improved.

Line 175, full name for abbreviation for BPCR.

Line 267, Line 271, Line 315, one or two-tailed test is to be stated.

Line 280, the sentence for Var(w) is to be revised. Var (w) refers to the variance of the standard logistic distribution used in the logistic link function.

Line 284, the PCV range and interpretation to be provided e.g. 0 < PCV < 1.

Line 286, more information for MOR to be provided e.g. quantify the unexplained cluster-level heterogeneity.

Line 369, typo p<000

Line 379, Line 391, Line 393, Line 396, R2, R squared, R-square is to be written as R^2 or R with 2 as superscript or R-squared. Need to be consistent/standardized.

Table 6 current age, the values o.83 and o.93 to be revised as 0.83 and 0.93 respectively.

References did not conform to the journal format.

7. PLOS authors have the option to publish the peer review history of their article (what does this mean?). If published, this will include your full peer review and any attached files.

Reviewer #2: No

---

## [Author Response · Author response to Decision Letter 2]

4 Jul 2024

A point-by-point response to editor and reviewers

Authors’ Response to Academic Editor

Dear: Amanuel Abajobir, PhD, Academic Editor, Plos One

We thank you for a thorough reading and constructive comments and suggestions on our manuscript and for the opportunity to revise and resubmit. We are pleased to submit the revised version of the manuscript titled “Geographical variation and predictors of missing essential newborn care during the immediate postpartum period in Ethiopia: Spatial and multilevel count analyses” for your consideration in the special collection of Plos One. The comments of the editors and the reviewers were highly insightful and enabled us to greatly improve the quality of our manuscript. In this revised manuscript we made substantial changes to address your concerns in a point-by-point response. We appreciate your time and look forward to your response and we are very keen to incorporate further comments, if any, for the betterment of the final manuscript.

On the following pages, you will find our responses to the comments and suggestions raised by the esteemed editor and reviewer. 

Sincerely, 

Aklilu Habte (MPH)(corresponding author)

aklilihabte57@gmail.com

Response to editors’ comment

1. Please remove any reference to any 'ethnicity' in the 'Classification' of the submission system.

Response: thank you so much for your suggestion. Accordingly, we removed it in the submission system during our submission to revised version of the manuscript.

END________________________________________

 THANK YOU!!!

Authors’ Response to Reviewer#2 

Comment 1: The manuscript requires major editorial editing throughout. The cited references, abbreviations etc need to be spaced out from the words and figures consistently across the entire manuscript, including the tables. Please refer to a PLOS journal article for an example.

Response: Dear Reviewer 2, thank you very much for taking the time to review our work and for your positive feedback. We received your thoughtful, and generous review, along with helpful feedback and suggestions, and we have corrected all the raised editorial issues throughout the revised version of the manuscript. For your convenience, we made a highlight to show the space between the reference and words.

Comment 2: The analysis conducted is comprehensive, but the manuscript could be further improved.

Response: we appreciate your concerns. However, we assure you that the whole content of the manuscript was the mirror image of the analysis. We performed descriptive, multilevel, and spatial analyses and all results were well presented accordingly.

Comment 3: Line 175, full name for the abbreviation for BPCR. 

Response: Thank you very much for your meticulous review. It was a typo error and we have corrected and highlighted it in the ‘Measurement of variables of the study’ section of the "Revised Manuscript with Track Changes" on page 6, Lines 174-175

Comment 4: Line 267, Line 271, Line 315, a one or two-tailed test is to be stated.

 Response: we appreciate your insightful inquiry. We performed an independent t-test and one-way Analysis of variance (ANOVA) and we have corrected and highlighted it in the ‘data analysis’ section of the "Revised Manuscript with Track Changes" on page 10, Line 267

Comment 5: Line 280, the sentence for Var(w) is to be revised. Var (w) refers to the variance of the standard logistic distribution used in the logistic link function.

Response: thank you for your meticulous review and insightful suggestion. We have corrected and highlighted it in the ‘Data analysis’ section of the "Revised Manuscript with Track Changes" on page 10, Lines 280-281

Comment 6: Line 284, the PCV range and interpretation to be provided e.g. 0 <PCV < 1.

Response: thank you for your suggestion. We have added and highlighted it in the revised version of the manuscript, Lines 285-286, Page 10 

Comment 7: Line 286, more information for MOR to be provided e.g. quantify the unexplained cluster-level heterogeneity.

Response: we have added the statement and highlighted it in the revised version of the manuscript, Lines 287-288, Page 10.

Comment 8: Line 369, typo p<000

Response: We have corrected and highlighted it on Line 371, Page 15

Comment 9: Line 379, Line 391, Line 393, Line 396, R2, R squared, R-square is to be written as R^2 or R with 2 as superscript or R-squared. Need to be consistent/standardized. 

Response: Thank you for your meticulous review. We have corrected it as R2 and highlighted it as per your suggestion. 

Comment 10: Table 6 current age, the values o.83 and o.93 to be revised as 0.83 and 0.93 respectively.

Response: we have corrected and highlighted it in Table 6 of the "Revised Manuscript with Track Changes" Page 19.

Comment 11: References did not conform to the journal format.

Response: Thank you so much for your suggestion. As the Plos One journal requirement is the Vancouver style we made citations accordingly. Beyond that, this is the reference style that we used during our prior publication experience at this journal.

Thank you for your constructive comments and suggestions, which we got as valuable input in the improvement of our manuscript. We received all of them as a valuable contribution to our ongoing work. 

END_______________________________________

 THANK YOU!!!

---

## [Editor Report · Decision Letter 3]

17 Jul 2024

Geographical variation and predictors of missing essential newborn care items during the immediate postpartum period in Ethiopia: Spatial and multilevel count analyses

PONE-D-23-29356R3

Dear Author(s),

We’re pleased to inform you that your manuscript has been judged scientifically suitable for publication and will be formally accepted for publication once it meets all outstanding technical requirements.

Kind regards,

Amanuel Abajobir, PhD

Academic Editor

PLOS ONE

---

## [Editor Report · Acceptance letter]

19 Jul 2024

PONE-D-23-29356R3 

PLOS ONE

Dear Dr. Hailegebireal, 

I'm pleased to inform you that your manuscript has been deemed suitable for publication in PLOS ONE. Congratulations! Your manuscript is now being handed over to our production team.

Kind regards, 

on behalf of

Dr Amanuel Abajobir 

Academic Editor

PLOS ONE